# Architectural control of metabolic plasticity in epithelial cancer cells

Maia Al-Masri[1,2], Karina Paliotti[1], Raymond Tran [3], Ruba Halaoui[1,2], Virginie Lelarge[1], Sudipa Chatterjee[1,2], Li-Ting Wang[1,2], Christopher Moraes [1,3] & Luke McCaffrey [1,2,4✉]

Metabolic plasticity enables cancer cells to switch between glycolysis and oxidative phosphorylation to adapt to changing conditions during cancer progression, whereas metabolic dependencies limit plasticity. To understand a role for the architectural environment in these processes we examined metabolic dependencies of cancer cells cultured in flat (2D) and organotypic (3D) environments. Here we show that cancer cells in flat cultures exist in a high energy state (oxidative phosphorylation), are glycolytic, and depend on glucose and glutamine for growth. In contrast, cells in organotypic culture exhibit lower energy and glycolysis, with extensive metabolic plasticity to maintain growth during glucose or amino acid deprivation. Expression of KRAS$^{G12V}$ in organotypic cells drives glucose dependence, however cells retain metabolic plasticity to glutamine deprivation. Finally, our data reveal that mechanical properties control metabolic plasticity, which correlates with canonical Wnt signaling. In summary, our work highlights that the architectural and mechanical properties influence cells to permit or restrict metabolic plasticity.

[1] Rosalind and Morris Goodman Cancer Research Centre, McGill University, Montreal, QC, Canada. [2] Division of Experimental Medicine, McGill University, Montreal, QC, Canada. [3] Department of Chemical Engineering, McGill University, Montreal, QC, Canada. [4] Gerald Bronfman Department of Oncology, McGill University, Montreal, QC, Canada. ✉email: luke.mccaffrey@mcgill.ca

Metabolic reprogramming is a hallmark of cancer and it is essential to understand metabolic dependencies of cancer cells that can be exploited therapeutically[1–3]. Proliferating cancer cells use both glycolysis and mitochondrial oxidative phosphorylation (OXPHOS), an effect first reported nearly 100 years ago by Otto Warburg[4–7]. More recently, it has become appreciated that cancer cells may display metabolic plasticity and switch between glycolysis and OXPHOS depending on microenvironmental context in different tissues[8,9]. Although oxidative glycolysis produces less ATP than OXPHOS, it is proposed to contribute by providing carbon for anabolic pathways to support cancer growth[6,10]. In addition, reduction of glucose-derived pyruvate to lactate maintains redox balance of NADH/NAD+ to prevent cellular reductive stress in metabolically active cells[4,11]. Amino acids are another major source of carbon, as well as nitrogen, for generating biomass in proliferating cells, and nonessential amino acids like glutamine or serine are frequently conditionally required for cancer cell growth[1,12,13]. Amino acids and glucose can feed into the tricarboxylic acid (TCA) cycle, which provides precursors of lipids, nucleotides, and amino acids necessary to generate the macromolecules required by proliferating cells[14].

Oncogenes and tumor suppressors function as cell intrinsic genetic factors that can stimulate or constrain metabolic functions of cancer cells[2,15]. For example, KRAS is mutated in ~25% of all human cancers, which produces a constitutively active form of the protein that impairs mitochondrial function and induces glucose-dependency in transformed cells[16–18]. Moreover, KRAS-driven cancer cells become "addicted" to glutamine and also require it for growth[19,20]. More recent data indicate that non-genetic properties, including mechanical cues from cell–cell and cell–matrix interactions, can also promote glycolysis[21–23].

Epithelial cells interact to form architecturally organized structures in tissues, with strict control of motility, survival, and proliferation signaling, each of which is altered during cancer progression[24–27]. The propagation of cells in flat (two-dimensional; 2D) cultures is technically simple and enables maintenance of long-term exponential growth. However, they lack architectural information that regulates many epithelial cell behaviors, which are thought to be more accurately recapitulated in three-dimensional (3D) cultures[28,29]. 3D culture is a general term that describes diverse culture methods that are related by ability of cells to more freely associate with the microenvironment on all sides. For example, suspension spheroids typically grow as solid structures that may have inter-cellular diffusion gradients[30], but lack contextual information from the mechanical microenvironment. In contrast, cells in organotypic cultures preserve mechanical cell–ECM interactions and organize into tissue-relevant architectures that regulate signaling and growth[28,31]. Recent studies have noted metabolic differences between 2D and 3D cultures[30,32–34]. However, the influence of the architectural and mechanical environments on metabolic dependency and plasticity remained to be established.

Here we report that metabolic plasticity can be controlled by the architectural environment. Epithelial cancer cells and non-transformed cells cultured in flat 2D contexts are dependent on glucose and glutamine for growth, whereas in organotypic 3D cultures they exhibit metabolic plasticity to maintain growth during nutrient deprivation. Underlying these differences are unique metabolic programs between flat and organotypic cultures. Cells in flat/2D culture are highly glycolytic, whereas cells in organotypic culture have enriched amino acid metabolism. Remarkably, cells in organotypic culture retain a reserve glycolytic capacity that is accessed during amino acid or mitochondrial stress to support growth. Finally, we show that metabolic plasticity is in part regulated by extracellular stiffness and Wnt signaling. Together this demonstrates that the architectural environment plays a key role in regulating metabolic dependency and adaptability of cancer cells.

## Results

**Glucose-addicted growth is influenced by the architectural environment.** To address how cell architecture influences the metabolic behavior of epithelial cells, we chose Caco-2 cells as an experimental model. Caco-2 are derived from intestinal adeno-carcinoma that grow as flat monolayers and establish apical-basal polarity in standard culture on plastic. When cultured in the presence of basement membrane extract (BME), a hydrogel rich in basement membrane components, Caco-2 cells develop into polarized hollow cysts (Fig. 1a, b). Importantly, both flat cultures and cysts generate a cellular monolayer, which affords the ability to directly compare flat (2D) and organotypic environments on metabolism, while avoiding potential confounding issues of diffusion gradients in solid spheroids that may influence metabolism. Many cancer cell lines are dependent on glucose as an energy source in the presence of oxygen. To address the requirement for glycolysis in the Caco-2 model, we cultured flat and organotypic cells in glucose-replete or glucose-free medium. Whereas flat cultures showed a marked dependence on glucose (80% reduction in growth in the absence of glucose) no growth difference was observed for organotypic cells cultured in glucose-replete or glucose-free medium (Fig. 1c–e; Supplementary Fig. S1a). Moreover, glucose was not required for cell polarization, lumen formation, or gross epithelial morphology of organotypic cultures (Fig. 1d, f).

To test the possibility that the extracellular matrix provided glucose or another soluble factor that could support organotypic cultures in glucose-depleted conditions, we examined several parameters. First, we measured the glucose concentration in glucose-free medium containing gel matrix and did not detect glucose (Supplementary Fig. S1b). Secondly, we cultured cells as flat monolayers in the presence of soluble gel matrix at the same overall concentration used in organotypic cultures. Under these conditions, the addition of soluble gel matrix was unable to support growth of cultures of Caco-2 cells in glucose-free medium (Supplementary Fig. S1c). Therefore, we conclude that glucose, or another soluble factor in the gel matrix, was not responsible for maintaining growth properties in Caco-2 organotypic cultures in glucose-free medium.

Since glycolysis has been associated with growth of cancer cells[4] we compared the basal growth rate of Caco-2 cells in flat and organotypic cultures. Whereas both displayed exponential growth characteristics, organotypic cells had a ~25% longer population doubling time (Fig. 1g). This indicates that although both flat and organotypic conditions support exponential proliferation, only cells in flat cultures are dependent on glucose for growth.

To determine if the difference in glucose dependency observed between flat and organotypic cultures was specific to Caco-2 cells or was a more general feature, we performed similar experiments using cells from lung (A549), ovarian (OV90), and breast (MCF7) cancers and non-transformed breast epithelial cells (MCF10A). All cell lines tested displayed a significantly higher dependency on glucose for growth in flat compared to organotypic environments (Supplementary Fig. S1d–k). Of note, these additional cell lines do not form monolayer cysts with a central lumen indicating that this effect can be independent of monolayer status. Therefore, we conclude that the architectural environment is associated with differences in glucose-dependent growth.

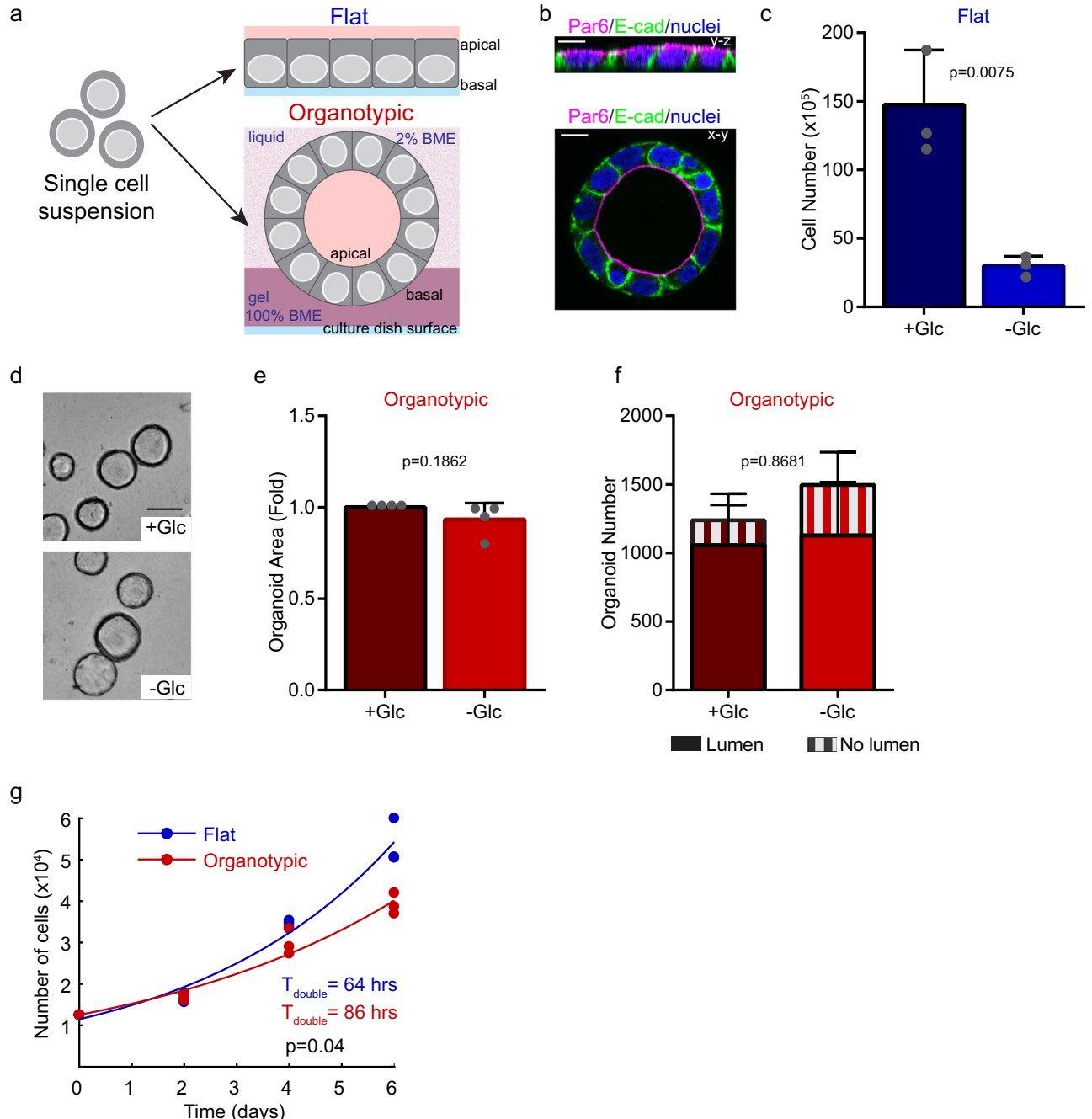

**Fig. 1 Epithelial architecture is associated with glucose-dependent growth. a** Diagram depicting experimental model whereby Caco-2 cells were seeded as flat or 3D organotypic monolayer cultures. Blue lines at the bottom represent the dish surface. For 3D organotypic cultures, cells were semi-embedded in gel matrix: Dark purple represents solid gel matrix (100% basement membrane extract, BME) and light purple represents liquid matrix (2% BME). **b** Fluorescent micrographs showing Caco-2 cells immuno-stained with E-cadherin (basolateral marker; green) and Par6 (apical marker; magenta) to indicate epithelial organization of the cell monolayers. **c** Bar chart showing quantification of Caco-2 cell numbers. Flat/2D cells were seeded and cultured for 6 days in the presence (+Glc) or absence (−Glc) of glucose ($r = 3$ independent replicates). **d** Brightfield images showing representative fields of Caco-2 organotypic cultures. Cells were cultured for 6 days in the presence (+Glc) or absence (−Glc) of glucose. Images show cysts with a hollow lumen, that appear as rings in cross-section images. Scale bars = 50 µm. **e** Bar chart showing quantification of Caco-2 cross-section area following 6-days of organotypic culture in the presence (+Glc) or absence (−Glc) of glucose ($n = 3715$ spheroids (+Glc), $n = 4195$ spheroids (−Glc)), ($r = 4$ independent replicates). **f** Bar chart showing morphology (lumen formation) of Caco-2 organotypic cells in the presence (+Glc) or absence (−Glc) of glucose (n=2476 (+Glc), $n = 2992$(−Glc)), ($r = 2$ independent replicates). **g** Scatter plots showing number of Caco-2 cells at 0, 2, 4, and 6 days of growth in flat or organotypic culture. Data were fit to exponential growth models (solid lines; $R^2_{flat} = 0.99$; $R^2_{organotypic} = 0.97$), ($r = 3$ independent replicates). All error bars are standard deviation.

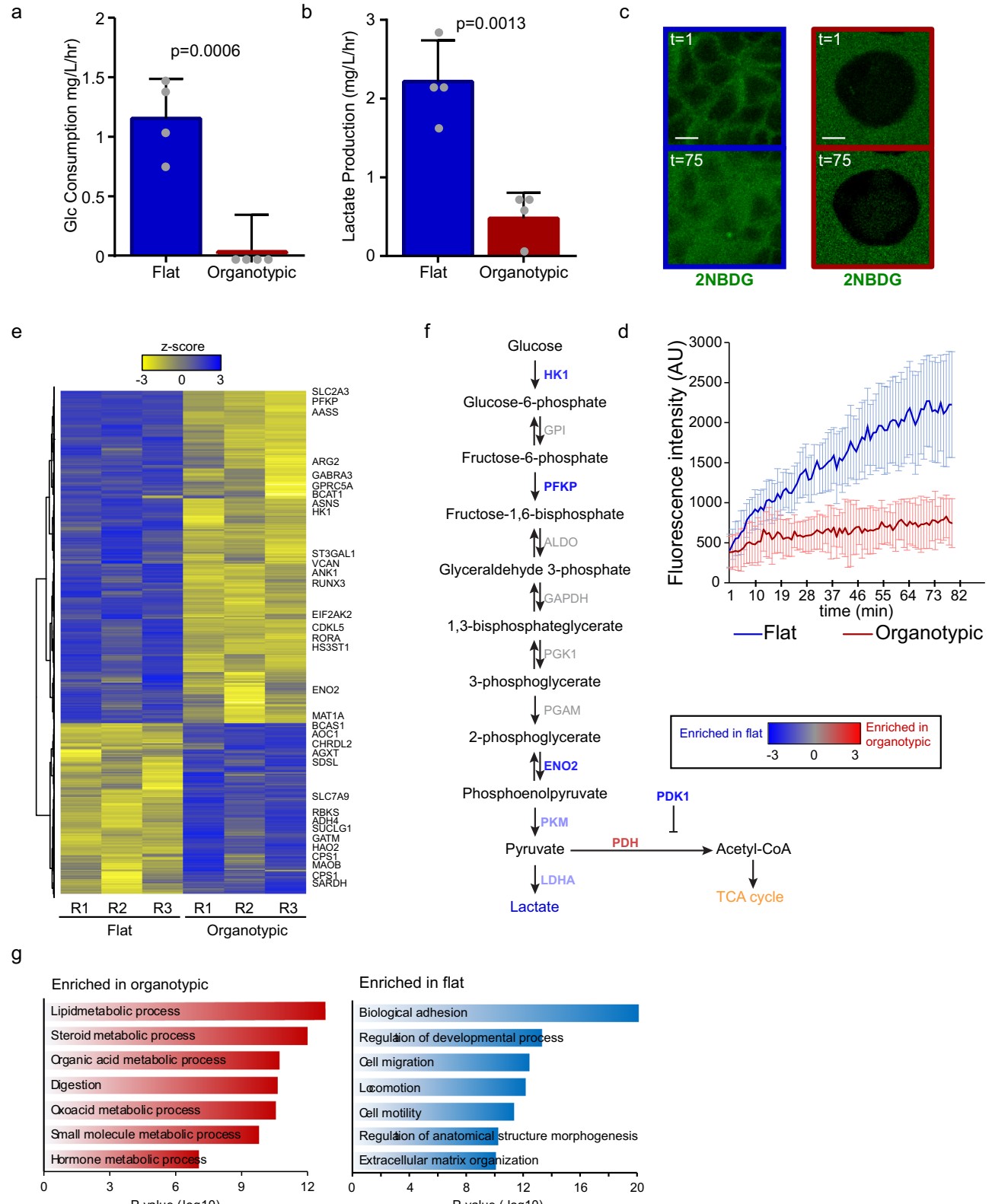

**Oxidative glycolysis is enhanced in flat cultures**. The observed differences in glucose dependence could reflect differences in glucose consumption or metabolic flexibility to nutrient stress. Strikingly, after 5 days in culture, we observed robust glucose consumption and lactate production for flat but not organotypic cultures (Fig. 2 a, b). Differences in glucose consumption were

validated using an independent kinetic assay measuring uptake of a fluorescent glucose analog FITC-2-NDBG (Fig. 2c, d).

To further explore differences in metabolic programs between flat and organotypic cultures, we performed RNA-sequencing in triplicate. We identified 1514 differentially expressed genes (False discovery rate (FDR) < 0.01, Supplementary Data 1), and flat and

**Fig. 2 Different epithelial architectures use unique glycolytic programs. a** Bar chart showing rate of glucose consumption from culture medium. Caco-2 cells were cultured for 6 days flat and organotypic cultures and the rate of glucose uptake (mg/L/h) was calculated. ($r = 4$ independent replicates). **b** Bar chart showing rate of lactate production from culture medium. Caco-2 cells were cultured for 6 days flat and organotypic cultures and the rate of lactate secretion (mg/L/h) was calculated. ($r = 4$ independent replicates). **c** Representative fluorescent images showing the glucose analog FITC-2-NBDG at initial ($t = 1$) and endpoint ($t = 75$). At the initial timepoints, dark regions are cells and the green show FITC-2-NBDG in the extracellular medium. Scale bars = 10 μm. **d** Line graph showing fluorescence uptake of the glucose analog FITC-2-NBDG for flat ($n = 78$ cells) and organotypic ($n = 38$ cells) Caco-2 cultures ($r = 2$ independent replicates). Cells were imaged every 2 min to observe and measurements made from intracellular regions. **e** Heatmap showing clustered RNA-seq data from 3 replicates (R1-R3) of Caco-2 cells from flat and organotypic cultures. Selected significantly altered metabolic genes (z-score threshold ±2) are displayed on the right. The full list of differentially expressed genes is available in Supplementary Data 1. **f** Pathway map for glycolysis showing differentially expressed genes and the reactions they catalyze. **g** Results of top-ranked Gene Ontology processes enriched in organotypic and flat Caco-2 cultures based on differentially expressed genes. All error bars are standard deviation.

organotypic groups were distinguished by hierarchal clustering (Fig. 2e). Gene Ontology analysis revealed that metabolic processes were the most highly enriched processes in organotypic cultures, whereas processes involved in adhesion and motility were enriched in flat cultures (Fig. 2f, g). Given the striking differences in glucose consumption and dependency between flat and organotypic cultures, we further explored the expression of glycolytic genes. Although there was no enrichment in glycolytic gene sets, we identified several key glycolytic genes that were expressed greater than 1.5-fold difference ($p < 0.05$; FDR < 0.01) in flat compared to organotypic cultures. Among enriched genes was hexokinase (HK1), which catalyzes the conversion of glucose to glucose-6-phosphate, the irreversible entrance into glycolysis, and PFKP, which catalyzes the conversion of fructose-6-phosphate to fructose-1,6-bisphosphate, the rate-limiting step of glycolysis (Fig. 2e, f, Supplementary Fig. S2a). Gene expression of glucose transporters SLC2A3 (GLUT3) and SLC2A1 (GLUT1) were also expressed at lower levels in 3D organotypic cultures compared to cells in flat cultures (Supplementary Fig. S2b). We also observed differences in expression of pyruvate kinase and lactate dehydrogenase, which are essential for the conversion of lactate to pyruvate as well as pyruvate dehydrogenase kinase (PDK1), which negatively regulates PDH and entry to the citric acid cycle to favor production of lactate (Fig. 2e, f). Overall, the gene expression data are consistent with the increased glycolysis we observed functionally in flat cultures and indicate that the architectural environment influences glucose consumption and the dependence on glycolysis for growth.

**Organotypic cells have lower energetics and higher reserve capacity than flat cells.** To assess the effect of the architectural environment on cellular energetics we simultaneously analyzed the extracellular acidification rate (ECAR) and cellular oxygen consumption rate (OCR) as measures of glycolytic activity and mitochondrial respiration, respectively. Furthermore, OCR measurements made in the presence of serially added oligomycin (inhibits ATP Synthase), FCCP (uncouples mitochondrial oxidative phosphorylation), and a mix of rotenone and antimycin A (inhibits electron transport chain), provides information on ATP-linked respiration, maximal respiration, and nonmitochondrial respiration, respectively[35]. Consistent with higher glycolysis in flat cultures, they also exhibited higher basal ECAR than cells in organotypic cultures (Fig. 3a, b). Flat cultures also exhibited higher basal OCR, maximal respiration, and ATP production indicating that flat cells are overall more energetic (Fig. 3c–e; Supplementary Fig. S3a, b). Increased metabolic rate correlates with proton leak[36], and consistent with this, we observed a threefold higher proton leak in flat cells (Supplementary Fig. S3c). Under conditions of mitochondrial energy stress, both flat and organotypic cultures increased ECAR (170% and 370% of baseline, respectively) indicating that both can increase glycolysis (Fig. 3a, b). However organotypic cells had significantly greater

relative glycolytic reserve, spare respiratory capacity, and coupling efficiency of oxidative phosphorylation (Fig. 3f, Supplementary Fig. S3d, e). The above results suggest that organotypic cells may increase glycolysis in response to oxidative phosphorylation stress. To confirm this, we treated cells with oligomycin and measured uptake of the fluorescent glucose analog, FITC-2-NBDG. Under these conditions, we observed a twofold increased dye uptake (Fig. 3g). Therefore, cells in organotypic cultures are characterized by low basal glycolysis and mitochondrial respiration, high metabolic reserve capacity, independence of glucose for growth, and metabolic plasticity.

**Flat and organotypic cells have different requirements for oxidative phosphorylation.** Mitochondrial respiration can be fueled by several substrates, including pyruvate, glutamine, and fatty acids[14]. Cytosolic pyruvate is transported into mitochondria via mitochondrial pyruvate carriers (MPC1/2), where it is converted to acetyl-CoA for entry into the TCA cycle to power the electron transport system[14]. Glutamine is converted to glutamate via glutaminase, which can be further oxidized to α-ketoglutarate to feed the TCA cycle[14]. Fatty acids are transported from the cytosol into mitochondria for fatty acid oxidation, which is dependent on carnitine palmitoyltransferase-1[14]. To test the contribution of diverse fuel sources for mitochondrial respiration we treated flat and organotypic cells with UK5099 (a mitochondrial pyruvate transporter inhibitor), BPTES (a selective glutaminase inhibitor), and Etomoxir, (an irreversible carnitine palmitoyltransferase-1 inhibitor) while monitoring OCR and ECAR. Following treatment of organotypic cells with UK5099, OCR was modestly reduced and ECAR augmented, whereas subsequent addition of BPTES and Etomoxir resulted in stronger effects (Fig. 3h, Supplementary Fig. S3g). In contrast, no effect on OCR, and a much smaller effect on ECAR, were observed for flat cells following these inhibitor treatments (Fig. 3h). To determine if glutamine and fatty acids are required for oxidative phosphorylation under basal conditions in organotypic cells, or are used as an adaptation to blocked pyruvate transporter, we added the inhibitors in the reverse order, BPTES and Etomoxir followed by UK5099. Under these conditions, we observed the strongest reduction in OCR only after inhibition of UK5099 in organotypic cultures (Fig. 3i). The inverse relationship between OCR and ECAR in organotypic cells supports our data that organotypic cells access a glycolytic reserve. This indicates that organotypic cultures can use multiple energy sources for mitochondrial respiration, and demonstrates metabolic plasticity.

**Flat and organotypic cells are differentially dependent on amino acids for growth.** After observing differences in glycolysis and mitochondrial respiration between flat and organotypic cells, we chose to further explore metabolic properties by profiling intracellular metabolites using GC/MS analysis

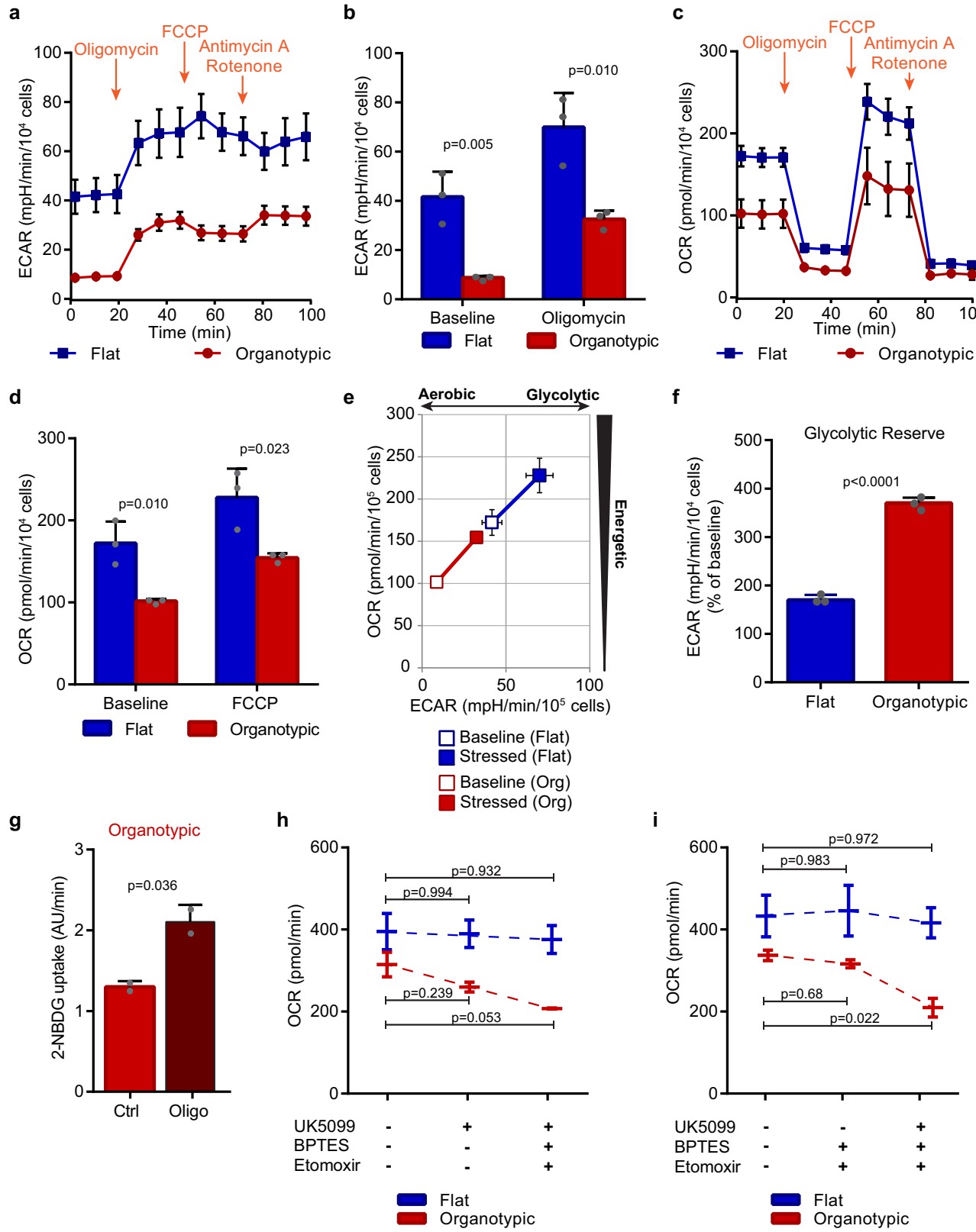

(Supplementary Data 2). Initial analysis of the retention tracings revealed substantial differences between flat and organotypic cultures (Fig. 4a). Quantification revealed significant enrichment in organotypic cells of several amino acids (glutamine, serine, methionine, phenylalanine, tyrosine, histidine, and threonine), the non-proteinaceous amino acid ornithine, and urea, a

product of amino acid metabolism (Fig. 4b and Supplementary Fig. S4a). Analysis of KEGG metabolic pathways revealed significant enrichment in organotypic cultures for pathways including Aminoacyl tRNA Biosynthesis, Glycine, Serine, and Threonine metabolism, Glyoxylate and Dicarboxylate Metabolism, Cysteine and Methionine Metabolism, and (Fig. 4c). To

**Fig. 3 Epithelial architecture is associated with energetic stress responses. a** Representative line graph showing extracellular acidification rate (ECAR) measurements from flat and organotypic Caco-2 cultures in response to sequential addition of oligomycin, FCCP, and Antimycin A with Rotenone. Each point is the mean of three measurements. **b** Bar chart showing ECARs for flat and organotypic Caco-2 cultures at baseline and following addition of oligomycin ($r = 3$ independent replicates). **c** Representative line graph showing tracing of oxygen consumption rate (OCR) measurements for flat and organotypic Caco-2 cultures in response to energetic stresses conferred by sequential addition of oligomycin, FCCP, and Antimycin A with Rotenone. Each point is the mean of three measurements. **d** Bar chart showing OCRs for flat and organotypic Caco-2 cultures at baseline and following addition of FCCP ($r = 3$ independent replicates). **e** Plot showing the bioenergetic profile of flat and organotypic Caco-2 cultures at baseline activity (open symbols) and in response to mitochondrial inhibition (closed symbols) ($r = 3$ independent replicates). **f** Bar chart showing the glycolytic reserve of cells in flat and organotypic Caco-2 cultures ($r = 3$ independent replicates). Glycolytic reserve was calculated from % difference in ECAR from baseline (time = 0–20) to oligomycin treated (time = 40–60). **g** Bar chart showing uptake rate of FITC-labeled 2-NBDG in control (Ctrl;) or cells treated with 1 μM oligomycin (Oligo) ($n = 30$ cells (Ctrl), $n = 30$ cells (Oligo)) from $r = 2$ independent replicates. Uptake rates were calculated from intracellular fluorescent intensity measurements from time-lapse images, and represent the slope. **h** Waterfall plot showing OCRs at baseline and following sequential addition of UK5099 followed by BPTES and Etomoxir for flat and organotypic Caco-2 cultures ($r = 2$ independent replicates). **i** Waterfall plot showing OCRs at baseline and following sequential addition of BPTES and Etomoxir followed by UK5099 for flat and organotypic Caco-2 cultures ($r = 2$ independent replicates). All error bars are standard deviation.

---

further understand different metabolic pathway utilization, we integrated metabolomics and gene expression data and mapped them to KEGG pathways (Supplementary Fig. S5). This revealed enrichment in organotypic cultures of interconnected metabolic pathways including the TCA cycle, urea cycles, and pyruvate biosynthesis (Supplementary Figs. S5). In addition, organotypic cells had increased levels of the TCA intermediate succinate, and expression of genes encoding enzymes that use succinate (Supplementary Figs. S4, S5).

The different intracellular levels of glutamine and other amino acids between flat and organotypic cells suggested to us that there may also be distinct requirements on amino acids for growth. Glutamine is a nonessential amino acid that is frequently required for cancer cell growth[1,19,20]. We therefore evaluated growth of flat and organotypic cultures in glutamine-containing or glutamine-free conditions. Whereas flat cells displayed a major growth deficiency in the absence of glutamine (30% of control), organotypic cells were substantially less dependent on glutamine for growth (85% of control) (Fig. 5a, b). To determine if the differential effect of glutamine dependence was a general feature of cells in flat and organotypic conditions, we also evaluated lung A549, ovarian OV90, and breast MCF7 cancer cells, as well as non-transformed MCF10A cells in glutamine-containing and glutamine-free conditions. In all cases, the cells were considerably less dependent on glutamine for growth in organotypic cultures (Supplementary Fig. S6a–h).

To determine if other amino acids with different intracellular levels between flat and organotypic cells also displayed distinctive growth requirements, we cultured cells in the presence or absence of the serine or methionine. In both cases, growth of flat cells was considerably reduced in the absence of serine or methionine (60% and <5% of control), whereas growth of organotypic cells was >90% of controls (Fig. 5c–f). To further examine metabolic dependency of cells, we cultured cells in a medium devoid of nonessential and essential amino acids. Remarkably, organotypic cells survived cultured in amino acid-free medium and displayed a 50% reduction in growth, whereas flat cells were unable to grow in these conditions (Fig. 5g, h). These data suggested to us that cells in organotypic cultures display greater plasticity than flat cells, with regards to nutritional dependency. We showed above that organotypic cells were not dependent on glycolysis but retained glycolytic capacity that could be accessed when mitochondrial respiration was blocked. We, therefore, hypothesized that glycolysis may help sustain proliferation in the absence of certain amino acids and tested this by co-depleting glucose and glutamine from the culture medium. Under these conditions, we observed a 40% reduction of growth in organotypic, compared to 15% reduction by glutamine alone (Fig. 5b, j). In all conditions

tested, there was no apparent effect on epithelial organization and lumen formation in organotypic cultures (Supplementary Fig. S6i). Therefore, we conclude that organotypic cells are partially dependent on glucose metabolism for growth under conditions of glutamine deprivation. Collectively, this supports that cells in organotypic cultures exhibit robust metabolic plasticity and have a glycolytic reserve that is engaged during amino acid nutrient stress.

**KRAS[G12V] induces glucose dependency in organoid cultures**. We next wondered if organotypic cells could be forced to become glucose-dependent for growth, and whether this affected their metabolic plasticity. Oncogenic transformation by KRAS[G12V] induces cells to become glycolytic[17], and KRAS-driven cancer cells also require glutamine for growth[19,20]. We established doxycycline-inducible KRAS[G12V] Caco-2 cells, which stimulates growth in both flat and organotypic cultures, and fill the lumen in organotypic cultures (Fig. 6a, b, Supplementary Figs. S7a–c). To confirm that KRAS[G12V] induces glucose dependency in Caco-2 cells, we cultured them in glucose-free medium with doxycycline, and observed severely restricted growth in both flat and organotypic formats (Fig. 6a, b; Supplementary Fig. S7d). Therefore, cells in organotypic culture retain susceptibility to oncogene-induced glucose dependency.

We next tested if KRAS[G12V]-expressing Caco-2 cells were also dependent on glutamine for growth. Control (GFP-expressing) flat cells grew to 30% of maximal growth in glutamine-free medium whereas KRAS[G12V]-expressing cells did not grow at all in the absence of glutamine (Fig. 6c), consistent with the requirement of glutamine in KRAS-driven cancer cells[19]. Control (GFP-expressing) organotypic cells grew to ~75% of maximal growth in glutamine-free conditions, which was exacerbated in KRAS[G12V]-expressing cells, which grew to ~30% of maximal growth in glutamine-free medium (Fig. 6d, e). Therefore, although organotypic KRAS-expressing cells have an increased requirement for glutamine compared to control cells, unlike flat cultures, they still retain a limited capacity for survival and growth when glutamine is absent from the culture medium.

To test if the requirement for other amino acids was also affected by KRAS[G12V] expression in organotypic cells, we cultured them in methionine-free or serine-free medium. Control (GFP-expressing) organotypic cells showed ~50%, and ~100% maximal growth in methionine-free and serine-free media, respectively. However, KRAS[G12V] organotypic cells showed ~30% maximal growth in methionine-free medium, and ~65% maximal growth in serine-free medium (Fig. 6d, e), indicating increased dependence on these nutrients. Therefore,

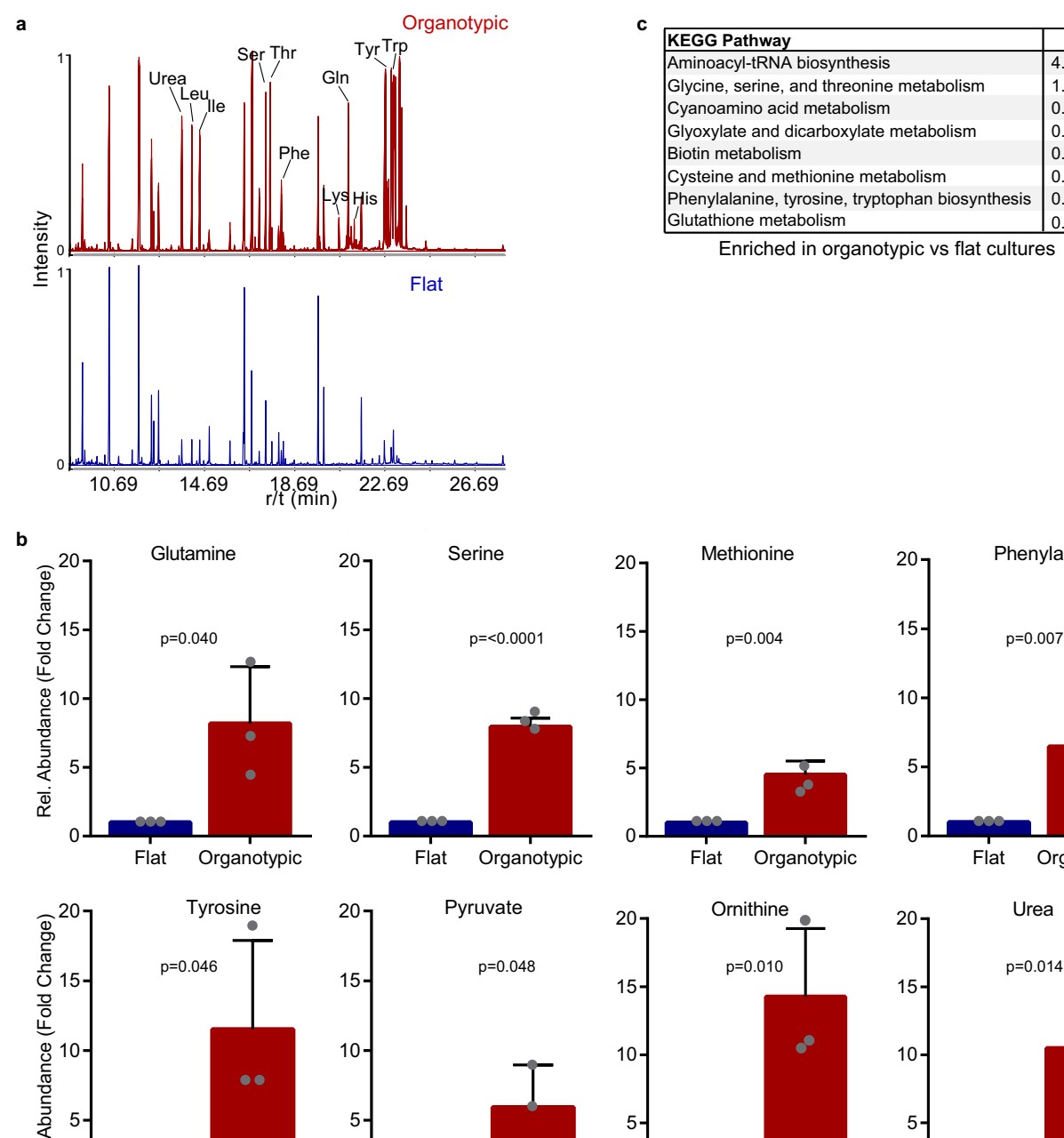

**Fig. 4 Amino acid metabolism is enriched in organotypic cultures. a** A representative GC/MS chromatogram of intracellular metabolites from flat and organotypic Caco-2 cultures. The data from metabolite measurements are supplied in Supplementary Data 2. **b** Bar graphs showing normalized levels of glutamine, serine, methionine, phenylalanine, tyrosine, pyruvate, ornithine, and urea from flat and organotypic Caco-2 cells. ($r = 5$ independent replicates). All error bars are standard deviation. **c** Table showing top ranked KEGG pathways enriched in organotypic versus flat Caco-2 cultures based on differentially expressed genes (Supplementary Data 1).

under conditions of KRAS$^{G12V}$-induced glycolysis, organotypic cells have reduced capacity to respond nutrient stress, but maintain some ability for survival and growth.

**The mechanical environment regulates glucose-dependent growth.** One major difference between flat and organotypic cultures is the mechanical microenvironment, whereby standard tissue culture dishes are much stiffer than ECM hydrogels (giga-Pascals (GPa) versus hundreds of Pa)[28]. To determine if the

mechanical environment influenced glucose-dependent growth, we cultured flat cells on polyacrylamide gels with a range of stiffness (shear modulus = 0.1–25 kPa) in glucose-containing or glucose-free conditions. On soft substrate (0.1 kPa), cell growth in the absence of glucose was significantly restored (50% of control). At greater stiffness, we observed a linear relationship (line fit: $y = -0.0007x + 0.577$; $R^2 = 0.999$) between glucose-dependent growth and stiffness between 0.1 and 0.8 kPa (Fig. 7a). On substrates stiffer than 0.8 kPa we observed a 90% reduction in growth

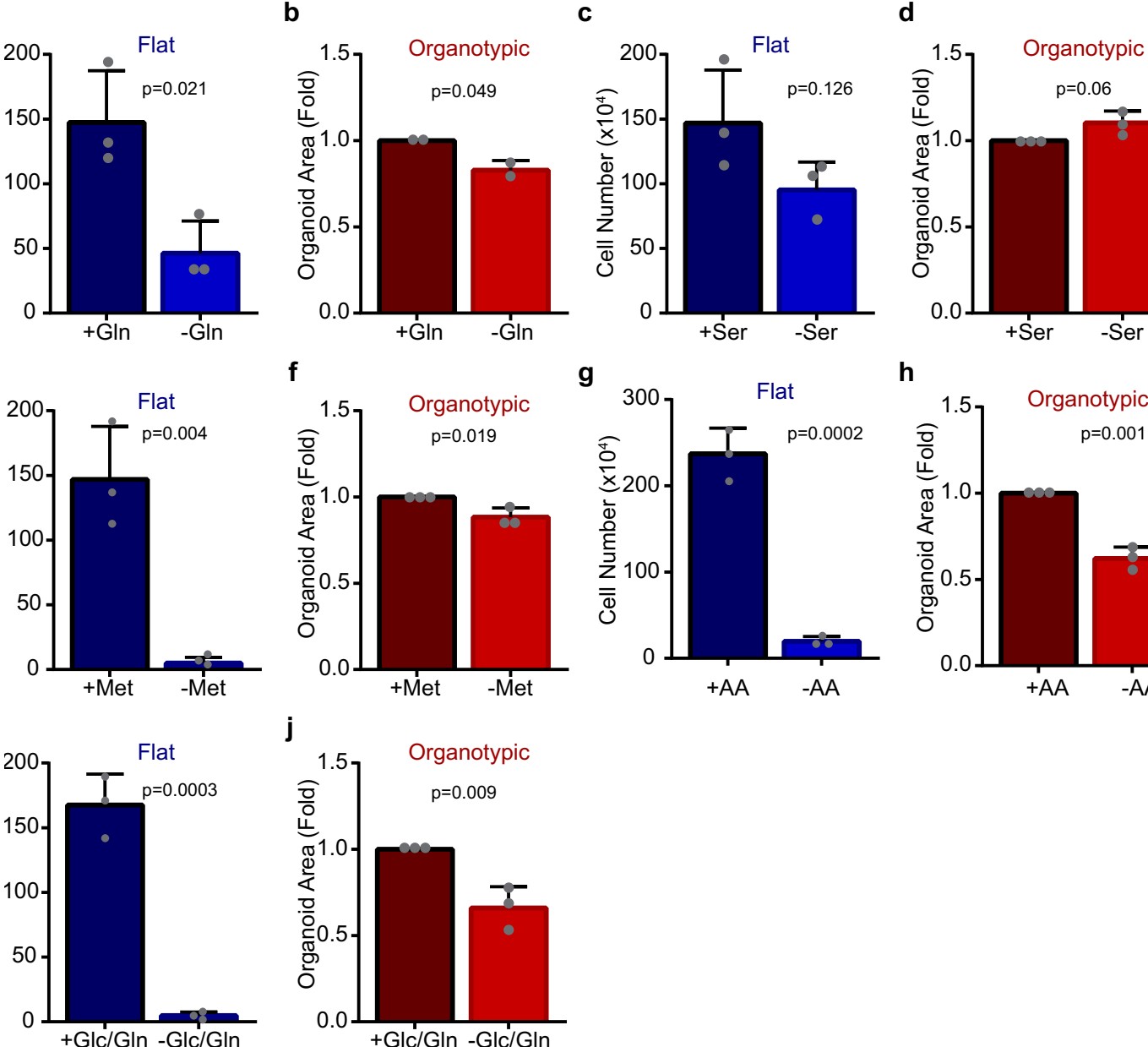

**Fig. 5 Organotypic cells exhibit plasticity to amino acid nutrient stress. a** Bar chart showing Caco-2 cell numbers. Caco-2 cells were counted after 6 days in flat/2D culture in the presence (+Gln) or absence (−Gln) of glutamine. ($r = 3$ independent replicates). **b** Bar chart showing cross-section area of Caco-2 spheroids. Caco-2 cells were grown in organotypic culture for 6 days in the presence (+Gln) or absence (−Gln) of glutamine, then the cross section size of spheroids was measured ($n = 2677$ spheroids (+Gln), $n = 2579$ spheroids (−Gln)), ($r = 2$ independent replicates). **c** Bar chart showing Caco-2 cell numbers after 6 days flat culture in the presence (+Ser) or absence (-Ser) of serine ($r = 3$ independent replicates). **d** Bar chart showing cross-section area of Caco-2 cell clusters after 6 days of organotypic culture in the presence (+Ser) or absence (−Ser) of serine ($n = 1926$ spheroids (+Ser), $n = 1957$ spheroids (−Ser)) ($r = 3$ independent replicates). **e** Bar chart showing Caco-2 cell number after 6 days of flat culture in the presence (+Met) or absence (−Met) of methionine ($r = 3$ independent replicates). **f** Bar chart showing size of Caco-2 cell spheroids after 6 days of organotypic culture in the presence (+Met) or absence (−Met) of methionine ($n = 1926$ spheroids (+Met), $n = 1941$ spheroids (−Met)) ($r = 3$ independent replicates). **g** Bar chart showing Caco-2 cell number after 6 days of flat culture in the presence (+AA) or absence (−AA) of essential and non-essential amino acids ($r = 3$ independent replicates). **h** Bar chart showing the size of Caco-2 spheroids after 6 days of organotypic culture in the presence (+AA) or absence (−AA) of essential and non-essential amino acids ($n = 2874$ spheroids (+AA), $n = 2541$ spheroids (–AA)) ($r = 3$ independent replicates). **i** Bar chart showing Caco-2 cell number after 6 days of flat culture in the presence (+Glc/Gln) or absence (−Glc/Gln) of glucose and glutamine ($r = 3$ independent replicates). **j** Bar chart showing size of Caco-2 spheroids after 6 days of organotypic culture in the presence (+Glc/Gln) or absence (−Glc/Gln) of glucose and glutamine ($n = 1951$ spheroids (+Glc/Gln), $n = 1446$ spheroids (−Glc/Gln)) ($r = 3$ independent replicates). All error bars are standard deviation.

relative to controls, similar to cells cultured on plastic (Fig. 7a). Since we observed differences in dependency on glutamine between flat and organotypic cells, we examined if this was also influenced by substrate stiffness. In the absence of glutamine, flat

cells were less dependent on glutamine for growth on soft substrates (0.1–0.4 kPa), and were more dependent on glutamine through a non-linear relationship with increasing stiffness (line fit: $y = -8.549e\text{-}6x + 0.417$; $R^2 = 0.21$) (Fig. 7a). To determine if

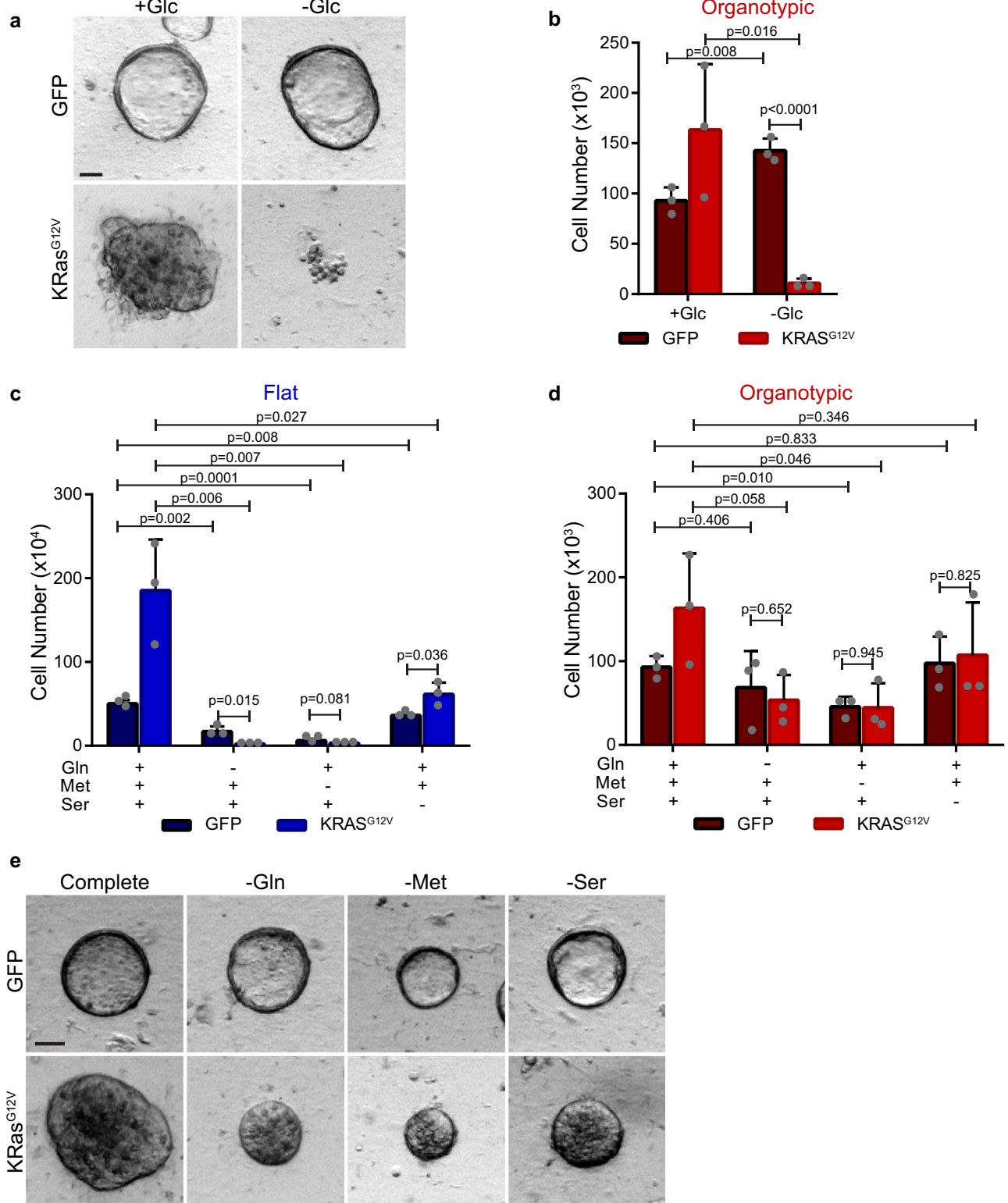

**Fig. 6 Organotypic cultures have reduced metabolic plasticity with Ras-driven glucose addiction. a** Representative brightfield images of control (GFP) or KRAS[G12V]-expressing Caco-2 cells after 6 days of organotypic culture in the presence (+Glc) or absence (−Glc) of glucose. GFP-expressing control cells display a hollow lumen, whereas KRAS[G12V]-expressing appear solid. Scale bars = 50 μm. **b** Bar chart of Caco-2 cell numbers depicted in organotypic cultures from **a**). Caco-2 cells were seeded in flat/2D or organotypic cultures and were counted after 6 days in culture. (*r* = 3 independent replicates). **c** Bar chart showing cell numbers of control (GFP) and KRAS[G12V] cells after 6 days of growth in the presence (+) or absence (−) of glutamine (Gln), methionine (Met), and serine (Ser). (*r* = 3 independent replicates). **d** Bar chart showing cell number from organotypic cultures of control (GFP) and KRAS[G12V] cells in the presence (+) or absence (−) of glutamine (Gln), methionine (Met), and serine (Ser) (*r* = 3 independent replicates). **e** Representative brightfield images showing organotypic control (GFP) and KRAS[G12V]-expressing cells in complete medium or medium free of glutamine (−Gln), methionine (−Met), or serine (−Ser). Scale bars = 50 μm. All error bars are standard deviation.

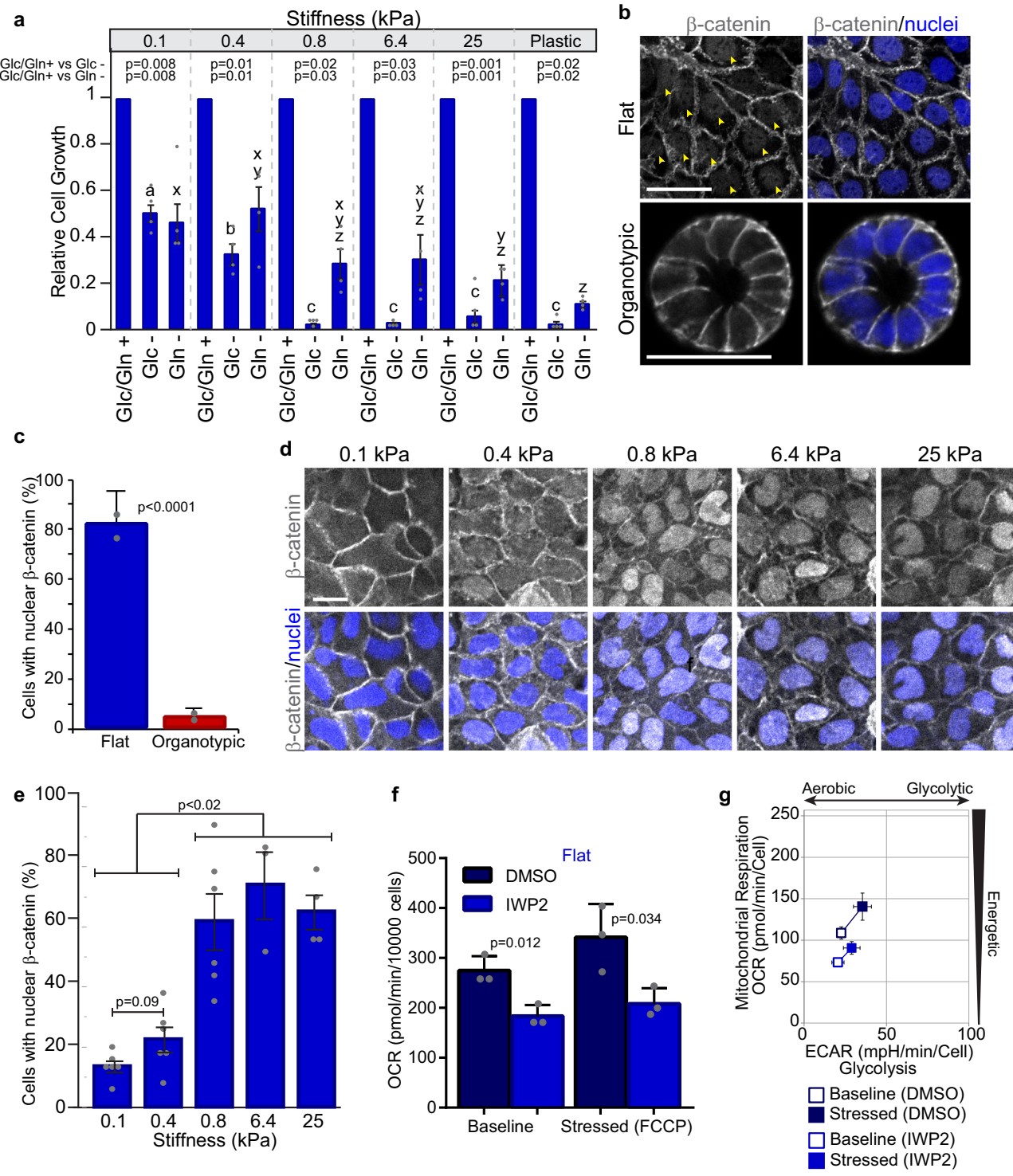

glucose and glutamine-dependency in 3D organotypic cells were also responsive to the mechanical environment, we fully embedded Caco-2 cells in BME with alginate. Alginate is an inert biopolymer that uses calcium as an ionic crosslinker to control the stiffness and is suitable for mechanical testing in 3D cell culture[37]. In gels containing 0.2% alginate, we observed an increased dependence on both glucose and glutamine (Supplementary Fig. S8a). Therefore, we conclude that the mechanical microenvironment influences nutrient-dependent growth and metabolic plasticity of epithelial cancer cells.

**Wnt signaling correlates with tissue stiffness and influences cellular energetics**. To understand signaling pathways that potentially underlie differences in flat and organotypic cultures, we mined our RNAseq data for pathways that were differentially regulated and identified components of the canonical Wnt pathways enriched in flat cultures (Supplementary Fig. S8b). To further investigate if Wnt signaling was different between flat and organotypic cultures, we immunostained for β-catenin, which localizes to the nucleus in canonical Wnt-stimulated cells[38]. In support of our gene-expression data, we observed a significantly

**Fig. 7 Extracellular stiffness contributes to different metabolic states between flat and organotypic cells. a** Bar chart relative growth of flat Caco-2 cells in complete (Glc/Gln+), glucose-depleted (Glc−), or glutamine-depleted (Gln−) medium. Cells were cultured for 5 days on polyacrylamide gels with shear moduli of 0.1, 0.4, 0.8, 6.4, and 25 kPa. *a, b, c*: $p < 0.05$ for different letters (Glc−); *x, y, z*: $p < 0.05$ for different letters (Gln−), ($r = 4$ independent replicates). **b** Fluorescence images of flat and organotypic Caco-2 cells after 5 days of culture immunostained for β-catenin and counterstained with Hoechst 33248 to label nuclei. Arrowheads show nuclear β-catenin. Scale Bars = 50 μm. **c** Bar chart showing the proportion of cells in flat and organotypic Caco-2 cultures with β-catenin detected in the nucleus. ($n = 20$ fields of view from $r = 2$ independent replicates) **d** Fluorescence images of flat Caco-2 cells immunostained for β-catenin and counterstained with Hoechst 33248 to label nuclei. Cells were seeded on various stiffness of polyacrylamide gels and cultured for 5 days prior to immunostaining. Scale Bars = 50 μm. **e** Bar chart showing the proportion of flat Caco-2 cells with nuclear β-catenin in cells depicted in **d**). ($n = 3$–6 fields of view from $r = 3$ independent replicates). **f** Bar chart showing OCRs for control (DMSO) or IWP2-treated (25 μM) flat Caco-2 cells at baseline and following addition of FCCP. Cells were cultured for 5 days, then IWP2 was added 24 h prior to measuring OCR. ($r = 3$ independent replicates). **g** Plot showing the bioenergetic profile of control or IWP2-treated (25 μM) flat Caco-2 cultures at baseline activity (open symbols) and in response to mitochondrial oxidative phosphorylation uncoupler, FCCP (closed symbols) ($r = 3$ independent replicates). All error bars are standard deviation.

higher proportion of cells with β-catenin localized to the nucleus in flat cells than organotypic cells (Fig. 7b, c). We next tested if nuclear β-catenin was dependent on stiffness in our flat culture model, and observed a mechanical-dependent increase in nuclear β-catenin (Fig. 7d, e). To determine if Wnt signaling may contribute to metabolic differences in flat and organotypic cultures, we treated cells with IWP2, a porcupine inhibitor that broadly blocks processing and secretion of Wnt proteins[39], then monitored effects on ECAR and OCR as readouts for glycolysis and mitochondrial respiration. While there was minimal effect on ECAR, the baseline OCR was reduced in flat IWP2-treated flat cells, which corresponded to a shift towards a lower energy state (Fig. 7f, g; Supplementary Fig. S8c–e). In contrast, IWP2-treatment did not have an effect on OCR, ECAR, or energy state in organotypic cultures (Supplementary Fig. S8f–i), consistent with lower canonical Wnt signaling in this condition. Therefore, these experiments demonstrate that a stiff 2D environment correlates with nuclear β-catenin, and that inhibition of the Wnt-regulator, porcupine, reduces the energetic state of these cells.

## Discussion

Here we comprehensively demonstrate that cells in 3D organotypic environments have striking differences in metabolic dependencies and plasticity compared to conventional flat 2D cultures. A key finding of our study is that cells in organotypic culture exhibit much greater metabolic plasticity. Relative to organotypic cells, cells in flat culture exist in an elevated energetic state, characterized by higher glucose consumption, lactate production, and oxygen consumption. One consequence of these differences is that flat cells are highly dependent on glucose, glutamine, and other nutrients for growth, whereas cells in organotypic culture can maintain exponential growth in the absence of supplemented glucose or glutamine.

Aerobic glycolysis is proposed to support exponential proliferation in part by supplying carbon for biosynthesis of macromolecules[6]. We observed a 25% shorter doubling time for cells in flat versus organotypic cultures. However, this was accompanied by a 400–500% increase in glycolysis, as measured by glucose consumption and lactate production. Consistent with our data, previous studies also report a non-linear relationship between proliferation rates and glycolysis[32,33,40–42]. This non-linear relationship indicates that glycolysis meets additional demands of cells, beyond biomass synthesis. One such function is to maintain the cytosolic NADH/NAD+ ratio, which is accomplished by the reduction of pyruvate to lactate to supply NAD+ necessary for multiple steps of the TCA cycle and other cellular functions[6,11,20,43]. This could explain, in part, our observations that cells in 2D flat cultures exhibit glucose-addicted growth and concurrent energetic mitochondrial respiration.

Amino acids are another major source of biomass in proliferating cells since they can supply both carbon and nitrogen[13,43]. Our data show that cells in 3D organotypic cultures can survive when essential and non-essential amino acids are excluded from the culture medium. Previous studies indicate that enhanced pinocytosis in Ras-driven cancer cells can promote uptake of extracellular soluble proteins to support amino acid metabolism during glutamine deprivation[44,45]. Since we cultured organotypic cells in protein-rich basement membrane extract, it is possible that this provides a source of soluble proteins that may be pinocytosed to support amino acid metabolism in 3D organotypic cultures. However, our data also demonstrate that access to a glycolytic reserve in organotypic environments restores growth under glutamine-deficient growth conditions, suggesting that diverse mechanisms for metabolic adaptability likely exist in tissues. Our metabolomic and RNA-Seq data indicate that amino acid pathways were enriched in organotypic cultures, including those that can generate pyruvate. This suggests that amino-acid-derived pyruvate may be more important in organotypic than flat culture conditions. Interestingly, it was recently shown that transamination of alanine to pyruvate functions can serve as a carbon source for the TCA cycle[12]. Functionally, we found that pyruvate, glutamine, and fatty acid pathways cooperated in organotypic cells to provide fuel for mitochondrial respiration. This supports the interpretation that cells in organotypic cultures are wired to utilize multiple energy sources that underlies plasticity.

What advantage might cancer cells have by maintaining lower basal levels of glycolysis? We speculate that although robust glycolysis supports a highly energetic state and a modestly increased growth, the trade-off is reduced metabolic plasticity. Under conditions of low basal glycolysis, we demonstrated that organotypic cells can access spare glycolytic capacity to readily adapt to conditions of mitochondrial stress or glutamine deprivation. In support of this trade-off, we observed that expression of KRAS[G12V] in organotypic cells induced their dependence on glucose, and reduced their ability to adapt to glutamine deprivation. In line with this we found that of all the cell lines we tested, lung A549 was most sensitive to glucose and glutamine withdrawal, and this is the only cell line tested that has a KRAS activating mutation. However, unlike flat cells, organotypic cells were still capable of minimal survival and proliferation, indicating that a degree of plasticity is preserved. We, therefore, suggest that the high glycolytic rate observed in flat cells contributes to a reduced capacity to adapt metabolically to changing nutrient demands.

Whereas it is well established that oncogenic drivers such as KRAS, MYC, and PI3K can induce metabolic reprogramming, our data indicate that cellular architecture and mechanical properties of the microenvironment are additional factors that

can impose metabolic dependency. Consistent with our observations, mechanical inputs are associated with increased glycolysis[21,46,47]. Here we demonstrate that both the architectural and mechanical environments are key modulators of metabolic plasticity and consequently affect metabolic dependencies. This may have important implications for cancer progression, which is associated with increased stiffness and changes to tissue organization[48]. We also found that this is accompanied by increased nuclear β-catenin, indicative of increased canonical Wnt signaling, and that inhibition of Wnt signaling reduces OCR in flat cultures. Together, these data suggest that Wnt signaling may contribute to the high energy state of flat 2D cell cultures. Indeed, nuclear β-catenin is responsive to mechanotransduction (our data and refs. [49–51]) and PDK1 was reported to be a Wnt-responsive gene[38]. Therefore, it is conceivable that Wnt could promote glycolysis through PDK1 by suppressing the entry of pyruvate to the TCA cycle in a mechanically-regulated fashion, a model that requires testing in the future.

Metabolic plasticity will likely limit the effectiveness of therapeutics that target metabolism directly or indirectly. Our findings that cell architecture sustains metabolic plasticity may help explain why metabolic inhibitors can have potent antiproliferative effects in cell lines in 2D culture, but show limited success for the same compounds in vivo and in clinical trials[52–54]. Indeed cellular responses to drug treatments are sensitive to the architectural environment, and cells cultured in 3D models tend to show more resistance to anticancer drugs than 2D cultures[30,33,55]. Our data demonstrating that cancer cells can exhibit profound glucose-independent growth may also help explain why glycolytic inhibitors are ineffective as monotherapy[43]. However, our experiments showing increased glucose sensitivity when other metabolic pathways are disrupted indicates that combination therapy targeting metabolic plasticity may hold greater promise.

Although our study uncovers an important relationship between the architectural environment and metabolic plasticity, several questions remain unanswered. Future studies will be needed to identify how interconnected metabolic pathways that confer plasticity are wired and regulated in diverse cancer contexts. Moreover, how do architectural and mechanical changes observed during cancer progression relate to adaptations in glycolysis and other metabolic dependencies?

In conclusion, there is a growing appreciation that the architectural environment influences myriad cellular processes, including proliferation, survival, morphogenesis, and chromosome segregation fidelity[29,56,57]. The paradigm uncovered here demonstrates that the architectural and mechanical environments also influence metabolic plasticity, which could have broad implications for our fundamental understanding of tissue development and cancer progression.

## Methods

**Cell culture**. Caco-2, A549, MCF7, OV90, and MCF10A cells were cultured for five or seven days in Dulbecco's Modified Eagle's Medium (DMEM 1X) (Wisent Cat# 319-005-CL) supplemented with 10% fetal bovine serum (FBS) (Wisent Cat# 080-150) and 5% penicillin–streptomycin solution (Wisent Cat# 450-201-EL). MCF10A cultures were further supplemented with 20 ng/ml EGF (Peprotech Cat# GMP100-15), 0.5 µg/ml Hydrocortsine (Sigma Cat#H-0888), 100 ng/ml Cholera Toxin (Sigma Cat# C-8052) and 10 µg/ml Insulin (Sigma Cat#I-1882). For organotypic cultures, single-cell suspensions were plated on a layer of 100% Geltrex™ LDEV-Free Reduced Growth Factor Basement Membrane Matrix (Gibco Cat# A1413202) and cultured in DMEM 1X (Wisent Cat# 319-005-CL) supplemented with 2% Geltrex™. This generated cells that are on top of a basement membrane substrate, while being exposed to soluble basement membrane components in a top liquid phase that supports epithelial organization and growth in 3D (Fig. 1a). Experiments with DMEM deprived of glucose or amino acids was supplemented with 10% dialyzed FBS (Wisent Cat# 080-910). Culture media used was amino acid-free

DMEM (Wisent Cat# 319-004-CL), L-serine-free DMEM (Wisent Cat# 319-017-CL), L-glutamine-free DMEM (Wisent Cat# 319-025-CL), L-methionine-free DMEM (Wisent Cat# 319-046-CL), D-glucose/pyruvate-free DMEM (Wisent Cat# 319-061-CL), and D-glucose/pyruvate/L-glutamine-free DMEM (Wisent Cat# 319-062-CL).

KRAS^G12V-expressing or GFP control Caco-2 cells were generated by lentiviral transduction using the doxycycline inducible lentiviral vector pCW. pCW-Cas9 was a gift from Eric Lander & David Sabatini (Addgene plasmid # 50661). pLenti-PGK-KRAS4B(G12V) was a gift from Daniel Haber (Addgene plasmid # 35633). Cas9 was replaced with FLAG-tagged KRAS^G12V or GFP using the NheI and BamHI restriction sites.

**Analysis of growth dynamics**. Proliferation was assessed by direct counting of viable cells by trypan blue exclusion with a hemocytometer following trypsinization (0.25% Trypsin; Wisent Cat# 325-043-EL). To determine the size of 3D organotypic structures, phase-contrast images were captured using an Axiozoom microscope (×56 zoom; Zeiss). The cross-sectional area of individual multicellular aggregates was measured using a custom macro in ImageJ/FIJI that segmented images based on thresholding and the Analyze Particles function.

**Fluorescent 2-NBDG uptake assay**. For kinetic experiments, cells were cultured in eight-well µ-slide chamber (Ibidi Cat# 80826) and incubated with FITC-labeled 2-(N-(7-nitrobenz-2-oxa-1,3-diazol-4-yl) amino)-2-deoxyglucose (2-NBDG) (Invitrogen Cat# N13195) for 1 h. A single image plane through the center of each cell (flat) or middle plane of the 3D structure was captured every 2 min for up to 2 h using a 20 × 0.8 NA objective lens at 16-bit depth using an LSM700 confocal microscope (Zeiss). Regions-of-Interest (ROIs) were manually selected within the cytoplasm of each cell and mean intensity was measured at each timepoint. The intensities were normalized to the initial intensity using the formula $I_{normalized} = I_{t=n} - I_{t=0}$, where $I$ is mean pixel intensity and $t$ is the time-point.

**Extracellular glucose and lactate measurements**. For quantification of extracellular glucose and lactate in the culture medium, Caco-2 cells were plated in 12-well dishes, in the presence or absence of Geltrex™ and incubated with DMEM for five days (~60% confluency for flat cells) prior to sample collection. Medium from flat or organotypic cultures was clarified by centrifugation and analyzed using the BioProfile 400 analyzer (Nova Biomedical Canada, Ltd. Mississauga, Ontario). Glucose consumption and lactate secretion rates were calculated as follows:

$$uptake/secretion\ rate = \frac{\Delta\ metabolite}{\Delta\ cell\ number} * growth\ rate$$

$$growth\ rate = \frac{\ln(cell\ number\ (T1)) - \ln(cell\ number\ (T0))}{time(T1) - time(T0)}$$

$$\Delta\ metabolite = [metabolite\ (final)] - [metabolite\ (blank)]$$

where blank wells contained medium, but no cells.

**Metabolic analyses**. The Seahorse XF Mito Stress test (Agilent Cat# 103015-100) and the Mito Fuel Flex test (Agilent, 103260-100) were performed using the XF24 Extracellular Flux Analyzer (Seahorse Bioscience) according to the manufacture's protocols. For both flat (2D) and organotypic (3D) cultures, cells were seeded in XF24 microplates (Agilent Cat# 100777-004). For organotypic cultures, wells were pre-coated with a thin layer of 100% Geltrex, and 2% Geltrex was added to the liquid culture medium, similar to standard growth conditions described above. Cells were cultured for 5 days, during which cells are in log-phase growth. At this timepoint flat/2D cultures form colonies and are <70% confluent; spheroids are ~50 µm and therefore readily fit under the probe, which is located 200 µm above the bottom of the well. Since cells in both 2D and 3D cultures are covered by liquid medium, special spheroid plates were not necessary for 3D cultures and the same XF24 microplates were used for all experimental conditions. On the day of the measurement, cells were washed with XF Base Medium (Seahorse Bioscience Cat# 102353-100) supplemented with D-glucose (25.0 mM) (Sigma Cat# G6152), L-glutamine (4.0 mM) (Bishop Cat# GLU102), and sodium pyruvate (1.0 mM) (Wisent Cat# 600-100-EL). Cell were then incubated in a $CO_2$-free incubator at 37 °C for 1 h to establish equilibration. For the Mito Stress test, measurements of the oxygen consumption rate (OCR) and extracellular acidification rate (ECAR), were taken before and after the addition of the ATP Synthase inhibitor oligomycin (1 µM), FCCP (0.5 µM) which uncouples oxygen consumption from ATP production, and Rotenone/antimycin A (0.5 µM), which inhibit Complex I and III respectively. For the Mito Fuel test, OCR and ECAR measurements, were taken before and after the addition of the mitochondrial pyruvate carrier inhibitor UK5099 (2 µM), carnitine palmitoyltransferase 1 A inhibitor etomoxir (4 µM), or glutaminase inhibitor BPTES (3 µM). For experiments comparing 2D and 3D environments, samples were run in parallel wells on the same plate. In both tests, following ECAR and OCR analyses, cells were trypsinized for cell count and data normalization. ECAR and OCR tracings, glycolytic reserve, maximum respiration, ATP production, proton leak, spare respiratory capacity, coupling efficiency, and non-mitochondrial respiration were calculated from OCR data using Seahorse Wave software (Seahorse Biosciences) and Excel (Microsoft). For both flat (2D)

and organotypic (3D), Caco-2 cells were treated for 24 h with the porcupine/WNT pathway inhibitor, IWP2 (25 µM) (STEMCELL Technologies Cat# 72122) prior to the procedures of the Seahorse XF Mito Stress test described above.

**RNA-seq analysis**. RNA was extracted from flat and organotypic Caco-2 cell cultures using the RNeasy kit (Qiagen Cat# 74004). RNA-seq was performed by Novogene (Beijing, China). Adapter sequences and low-quality score bases (Phred score < 30) were first trimmed using Trimmomatic[58]. The resulting reads were aligned to the human genome reference sequence (GRCh38/hg38), using STAR[59]. Read counts were obtained using HTSeq[60] and are represented as a table which reports, for each sample (columns), the number of reads mapped to a given gene (rows). For all downstream analyses, we excluded lowly-expressed genes with an average read count lower than ten in all of the samples, resulting in 15,945 genes in total. The R package limma was used to identify differences in expression levels between noninfected and infected samples at each time point. Nominal p values were corrected for multiple testing using the Benjamini-Hochberg method. The complete list of differentially expressed genes can be found in Supplementary Data 1.

**GC/MS metabolic analysis**. Metabolic profiling was performed as previously described[61]. Briefly, cells were cultured in DMEM in six-well dishes (35 mm) for 5 days. Cells were rinsed in saline, quenched in 80% HPLC-grade methanol, sonicated, centrifuged and supernatants were dried in a cold trap (Labconco) overnight at −1 °C. Pellets were solubilized in methoxyamine HCl, incubated at room temperature for 1 h and derivatized with MTBSTFA at 70 °C for 1 h. Next, 1 µL was injected into an Agilent 5975C GC/MS in SCAN mode and analyzed using Masshunter software (Agilent Technologies). Raw GC/MS data were converted to CDF format using Agilent MassHunter Workstation software with the following parameters: Max RT difference between peaks: 0.1 min and Ionization Type: EI. Metabolites were identified by automated comparison of the ion features in the experimental samples to a reference library of chemical standard entries that included retention time and molecular weight ($m/z$). Peak intensities were quantified using area-under-the-curve (AUC) for all samples. The mean AUC of each metabolite was calculated from five technical replicates to yield a single data point for each metabolite in flat/2D and organotypic/3D samples. The fold change was calculated as:

$$fold\ change = \frac{\overline{AUC_{3D}}}{\overline{AUC_{2D}}}$$

This was repeated for three independent experimental replicates and plotted as mean ± standard deviation.

The Joint Pathway Analysis module was used to assess Gene Ontology (GO) and Kyoto Encyclopedia of Genes and Genomes (KEGG) pathways enrichment against differential metabolites and genes (obtained from the RNA-seq analysis). Gene Set Enrichment Analysis (GSEA) was performed on a pre-ranked gene list (Supplementary Data 1) and the KEGG_GLYCOLYSIS_GLUCONEOGENESIS gene set using GSEA 4.1.0 software (Broad Institute). Leading edge analysis was performed with the same software to identify significant glycolytic genes upregulated in flat cultures.

**Stiffness analysis on polyacrylamide gels**. Cell culture compatible polyacrylamide (PAAm) gels were fabricated as previously described[62]. First, 18 mm glass coverslips were silanized in a solution of 3-(trimethoxysilyl) propyl methacrylate (0.4 % v/v in acetone) (Sigma Aldrich Cat# 440159) for 5 min followed by a 5 min rinse in acetone. PAAm prepolymer solutions were formulated using 40% acrylamide (BIORAD Cat# 161-0140), 2% bisacrylamide (BIORAD Cat# 161-0142), tetramethylethylenediamine (BIORAD Cat# T9281-100ML), 1% ammonium persulfate (APS; Fisher Cat# BP179-25), and phosphate buffered saline (PBS) to generate various experimental stiffness (shear modulus 0.1, 0.4, 0.8, 6.4, or 25.6 kPa). Polyacrylamide hydrogel stiffness was previously characterized using rheometry[62]. The gels were rinsed thoroughly with PBS for at least 24 h. The gels were submerged in a 0.1 mg/mL SulfoSANPAH solution and placed under UV light for 4 min and this step repeated twice. The PAAm gels were washed with PBS and then incubated overnight with a collagen I solution (1.5 mg/mL in PBS) (Advanced Biomatrix Cat# 5005-100ML) and then rinsed once with PBS before cell culture. Coverslips were transferred to 12-well dishes and 50,000 Caco-2 cells were plated and cultured as described above.

To analyze cell growth, coverslips were fixed in 4% paraformaldehyde for 20 min and stained with Hoechst 33258 (Fisher Scientific Cat# H1398) and images were captured using an Axiozoom microscope (Zeiss) at ×20 zoom. Analysis of cell coverage of each well was analyzed using custom macro in ImageJ/FIJI that thresholded the image based on Hoechst fluorescence and calculated the area covered by cells in each well.

**Alginate-basement membrane extract gels**. Sodium alginate powder (Sigma-Aldrich) was dissolved in phosphate-buffered saline (PBS) without Ca and Mg at 2% (w/v) and filtered through a 0.22 µm filter. 100% Geltrex™ LDEV-Free Reduced Growth Factor Basement Membrane Matrix (Gibco Cat# A1413202), Caco-2 cell

culture in DMEM 1X (Wisent Cat# 319-005-CL) to embed the cells, alginate stock solution was mixed to yield final ratios by volume of [50% Matrigel + 0.2% alginate stock]. 30 µL of hydrogel mixture was added per well followed by incubation at 37 °C for 1 h. CaCl₂ was dissolved in PBS without Ca and Mg at 1% (w/v) and filtered, and then diluted in media to a final concentration of 0.1%. [200] µL of CaCl2 media mixture was then added per well followed by incubation at 37 °C for 30 min. Organoids were then cultured in DMEM deprived of glucose or amino acids as described above.

**Immunofluorescence staining and western blot**. For immunofluorescence, cells were fixed in 4% paraformaldehyde for 20 min, blocked in 5% goat serum for 1 h, and incubated overnight with the followed primary antibodies at 4 °C: Par6B (1:100, Santa Cruz, sc-67393), E-cadherin (1:250, BD Transduction, 610181), β-catenin, (1:100, Cell Signaling Technology, D10A8), followed by the secondary antibody AlexaFluor 488 (Jackson ImmunoResearch, Cat# 111-545-144) at a 1/750 dilution for 1 h at room temperature. Imaging was performed using a LSM700 from Zeiss with 20×/0.8 NA or 40×/1.4 NA lenses. Image processing of brightness and contrast was performed in ImageJ/FIJI and was applied uniformly to the whole image.

For immunoblots, Caco2 cells were washed twice with ice-cod PBS and lysed in RIPA buffer (50 mM Tris-HCl, pH 8, 0.15 M NaCl, 0.1% SDS, 1% NP-40, 1% sodium deoxycholate, 50 mM NaF, 5 mM orthovanadate, 1 mM DTT) with proteinase inhibitor cocktail (Sigma # 11836170001). Cell lysates were centrifuged at 13,000 rpm at 4 °C for 20 min, and the supernatant was collected. Total proteins were denatured by SDS sample buffer and boiling in water for 5 min. The total proteins were separated by SDS-PAGE and transferred to Nitrocellulose membrane (Bio-rad # 1620115). Membranes were blocked in Tris-buffered saline containing 5% milk and 0.1% Tween 20. The primary antibodies and secondary horseradish peroxidase antibodies were used. Blots were washed with TBST three times and developed with the ECL system (Bio-Rad) according to the manufacturer's protocols. The primary antibodies were used as follows: rabbit anti-FLAG (1:1000; Delta Biolabs Cat# DB125), and mouse anti-tubulin (1:5000; Sigma, clone DM1A).

**Statistics and reproducibility**. Statistics were determined using GraphPad Prism, and JMP14.0.0 software. Results are presented as mean ± standard deviation. Comparison of two unpaired independent means was performed using a two-tail Student's $t$ test. Comparisons of multiple means were performed by ANOVA using Tukey's posthoc test. Growth curves were fit to an exponential equation using the curve fitting function in MATLAB. Data reproducibility is indicated as the number of independent experimental replicates (r) and the number organoids or technical replicates (n).

**Reporting summary**. Further information on research design is available in the Nature Research Reporting Summary linked to this article.

## Data availability
The authors declare that data supporting the findings of this study are available within the paper and its supplementary information files. Supplementary Data 3 contains the source data for the graphs and charts in the main figures. Any remaining information can be obtained from the corresponding author upon reasonable request.

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

## Acknowledgements

We thank the Rosalind and Morris Goodman Cancer Research Centre Metabolomics Core for technical assistance. We thank Alain Pacis for bioinformatics assistance. M.A.M. was supported by D'Avirro Family Foundation and Karrassik Foundation. R.H. was supported by the FRQS studentship. L.M. is a FRQS Research Scholar. This work was supported by a CIHR grant (PJT-156271) to L.M.

## Author contributions

Conceptualization, M.A.M. and L.M.; methodology, M.A.M. and R.T.; validation, M.A.M.; formal analysis, M.A.M., and L.M.; investigation, M.A.M., K.P., R.H., and S.J.C.; resources, R.T., C.M., V.L., and L.T.W.; data curation, M.A.M. and L.M.; writing—original draft, M.A.M. and L.M., writing—review and editing, M.A.M., L.M., K.P., R.T., C.M., R.H., S.J.C, V.L., and L.T.W.; Visualization, M.A.M., and L.M.; supervision, L.M., C.M.; funding acquisition, L.M.

## Competing interests

The authors declare no competing interests.
