## [Peer Review File · Communications Biology]

Reviewers' comments:

Reviewer #1 (Remarks to the Author):

In this article the authors compare the metabolic activity of cancer cells in 2D vs 3D. They found that cells cultivated in 2D are highly glycolytic and dependent on glucose and glutamine for growth, while cells cultivated on 3D have extensive metabolic plasticity, and thus are able to maintain growth even in glucose or amino acid depleted environment. Therefore, they investigate parameters that can influence the metabolic plasticity. They found that the metabolic plasticity of cells at least in part rely on their mechanical environment. Finally, in a not really convincing way (see below) they found some correlation between Wnt signaling pathway activation and stiffness-dependent metabolic rewiring

This work is interesting and pretty well executed, except for the last part (Wnt signaling pathway, see below). While it is of interest, the majority of the findings are confirmation of previous works. Nevertheless, some interesting findings should be investigating in depth to increase the novelty of their findings. Finally, experimental procedure of some experiments (Seahorse, metabolomics...) should be clarified.

Summary. This is an interesting article that continue to explore some important work started in 2009 by Schafer ZT et al, Nature 2009 that explored for the first time how loss of attachment to the ECM affects glucose and energy metabolism and more recent works (Bertero et al. J Clin Invest 2016, Bertero et al Cell Metabo 2019; Papalazarou et al Nat Metabo 2020, Park, J.S. et al Nature 2020; Romani et al., 2019) that investigate the connection between tissue/cell mechanics and cell metabolism. However, the study appears to be incomplete in potentially important ways. For this reason, I regretfully cannot recommend it for publication in COMMSBIO at this time.

Several shortcomings will need to be addressed before this research article is ready for publication.

1) The authors investigate whether ECM stiffness influence glucose and glutamine dependence of cancer cells in 2D. However, they only test 0.1kPa, 0.4kPa and 25kPa. They already found some differences between 0.1 and 0.4 kPa (both really soft ECM). It will be interesting to determine whether the effect of ECM stiffness on cell metabolism is linear or depend on a threshold? Also it will be of interest to determine whether ECM stiffness can also rewire glucose and/or amino acid metabolism in 3D cell culture.

2) As stated by the authors, cancer cells are known to be highly glycolytic compared to non-transformed cells. However, the authors shown that it is strongly dependent on how the cells are cultivated (2D vs 3D). Therefore, it will be of interest to determine: - whether non transformed cells present the same phenotype? - Whether cancer cells still present higher glycolytic rate than non-transformed cells in 3D cell culture?

3) To investigate metabolic changes the authors performed Seahorse experiments in 2 cell culture, but also in 3D cell culture. An extended experimental procedure describing how measure OCR and ECAR in 3D cell culture using Seahorse should be provided. At least for this reviewer it appears really hard to understand the technical procedure. Numerous technical caveats have to be lift (really small distance between the plate and the probe; non confluent area...)

4) The authors performed a GC/MS Metabolic Analysis to compare the metabolic profile of 2D vs 3D cell cultures. It is difficult to understand/interpret their results as the method of normalization remain enigmatic. Also the authors do not provide the full list of metabolites analyzed. Finally, in open science era, it will be of interest that the authors share their raw data.

5) The uptake rates are wrongly calculated as the authors state "Glucose consumption was calculated using the formula $\text{Consumption}_G = [\text{Glc}]_{\text{sample}} - [\text{Glc}]_{\text{blank}}$, where blank wells contained medium, but no cells. Values were then normalized to cell number in each well." This calculation is only correct in a no growth condition, but the author state their cells grow exponentially. For exponential growth, the calculation should read:

(The equation has been sent to the editor in a pdf file. Please confer with the editor to recover the pdf file.)

Please note Δ refers to the difference of a parameter between two time points.

6) Finally, guided by their RNA sequencing analyses the authors investigate whther Wnt signling can influence stiffness-dependent metabolic rewiring. The data presented here are only correlative and only rely on the use of a pharmacological inhibitor. Much more work (LOF/GOF and rescue

experiments) have to be done to demonstrate the involvement of Wnt signalling pathway in this process.

Reviewer #2 (Remarks to the Author):

In the manuscript "Architectural Control of Metabolic Plasticity in Epithelial Cancer Cells" the authors are comparing metabolic activities of CaCo-2 cell cultures on flat, stiff 2D substrates with organotypic cultures in soft matrix. They find that 2D cultures are more energetically active, consume more glucose, release more lactate, also use more oxygen and proliferate faster. In turn, the organotypic cultures exhibited a more active amino acid metabolism. When cells were stress with omission of individual amino acids from the culture medium, the organotypic cultures were less affected than 2D cultures. Next, they tested whether organotypic cultures could get addicted to KRAS G12V induced glycolysis, and they did. While even in the presence of oncogenic KRAS organotypic cultures tolerated absence of amino acids better than cells in 2D. They show that the glucose and glutamine requirements increased the stiffer the substrates were, suggesting that metabolic needs correlated with mechanical signaling. Finally, RNAseq experiments suggest that Wnt signaling pathway components were increased in flat cultures and indeed they showed more nuclear beta-catenin and both oxygen consumption and lactate secretion were sensitive to a Wnt inhibitor. Together, the data describe how metabolic activities and needs change when cells transition from a flat monolayer to the organotypic culture.

However, many open questions remain on how glucose and amino acid metabolisms are regulated under the different culturing conditions and how it mechanistically links to the presence of oncogenes. Still, I find the general observation that the metabolism of organotypic cultures showed more plasticity toward nutrient stress really interesting. Under these culture conditions cells seem to be able to adjust their metabolic requirements more readily than in 2D. This study may also suggest potential vulnerabilities of cancer cells to metabolic stress in the presence of reduced metabolic plasticity.

General comments

The manuscript is clearly written and structure and it is accessible. The figure legends tend to contain only a minimum of information. The readability of the figures would be enhanced if some more details were included. A couple more specific comments below relate to this. Some detail is missing from the methods section, a few more specific comments are included below. For some of the equipment that was used, the authors provide only minimal information. Details on important instrument settings are missing.

Specific comments

CaCo-2 cells were derived from colon adenocarcinoma and harbor mutations in APC and p53 and other genes, but not the MAP-kinase pathway. It was shown that p53 has inhibitory effects on glycolysis through TIGAR and glucose transporters. In this study glycolysis was only enhanced in the 2-dimensional culture but not in the organotypic cultures. To what extent would p53 deletion be effective in the context of 2D cultures versus 3D spheroids?

For the Caco-2 cells stably expressing GFP-KRAS G12V, evidence should be shown to what extent the fusion protein was over-expressed.

Regarding the fluorescent glucose uptake experiment shown in figure 2, example images for start and endpoints for both conditions should be provided to support the quantification data. Since multiple cell lines were used, the readability of the figures would be enhanced if the different cell lines were clearly indicated per panel.

Line 130: FDR should be written out the first time it is used.

Can the authors comment why in the gene ontology analysis glycolysis didn't show up as enhanced

pathway?

Figure 2F: I only found SLC2A3/GLUT3 labeled, were other glucose transporter gene expression levels analyzed as well?

Line 286: in this sentence, the 90% reduction of growth refers to glucose free conditions, I assume.

Line 351: the first sentence is incomplete.

Line 431: this sentence is broken.

Line 438: incomplete sentence.

Figure legends: it is not always clear what n and r refer to. Small r probably means "repeats" and should be spelled out at least once. For small n, it should indicate whether it refers to cells, organoids or other.

Legend panel 1A: since this panel is introducing the experimental setup, it would be helpful for the readers to know what percentage of matrix was used for solid and liquid phases.

Line 696: E-cad need to be spelled out. Along the same line, the information on antibodies for E-cadherin, Par6 should to be included in the methods section.

Line 510: this reference was not included

Scale bars: for panels where all images are displayed at the same scale a single scale bar is sufficient.

What is a organotypic culture with no lumen? Is it just small groups of cells? From panel 1d it appears that maybe one fourth of the structures had no lumen. What fraction of cells would this affect? I expect that in the organoids without lumen, the cells were not polarized. How would that impact the conclusion that it is the epithelial architecture that explains the metabolic differences?

In the abstract the authors state that cell lines from multiple cancers were analyzed, which they did. However, the main figures focus on Caco-2 cells and the data on the additional cell lines are found in the supplementary material. Moreover, while there are high resolution images of Caco-2 organotypic cultures provided and the cell polarization is shown by immunofluorescence staining, this analysis is missing for the other cell lines. So, it is not clear to what extent these cultures were polarized. I wonder whether it would be more suitable to focus the abstract on Caco-2 cells rather than include various tissues.

The Geltrex matrix is a composition of basement membrane proteins including collagen and laminin. Could the increased amino acid metabolism and part of the plasticity towards amino acid deprivation be explained by the presence of abundant matrix proteins that could potentially be processed by the cell?

Reviewer #3 (Remarks to the Author):

Overall comments: The authors attempted to demonstrate the differential behaviour of cancer cells cultured in a 2D monolayer and a 3D ECM scaffold. The marked differences were observed in these two cell culture models, the 2D model cell required glycolysis and mitochondrial activity compare to the 3D cell. This observation is consistently supported by several previous studies. The manuscript was well presented and written. There were several techniques employed by the author to prove the hypotheses. I am pleased to see how the author explain results and discussions in a non-complicated approach; it is easy to read and I really enjoyed reading the manuscript.

Minor comment: Line 280 "The mechanical environment regulates glucose – dependent growth". The author cultured cells on polyacrylamide gels which coated with collagen solution. If you want to prove your hypothesis of substrate stiffness could influence the glucose and glutamine dependency, why don't you use different concentration of Geltrex, which it was used to culture cells?

We thank the reviewers for their careful reading of our manuscript and recommendations. We address each of the reviewer comments and concerns point-by-point below (reviewer comments italicized and our responses are in bold).

Reviewer #1:

1. The authors investigate whether ECM stiffness influence glucose and glutamine dependence of cancer cells in 2D. However, they only test 0.1kPa, 0.4kPa and 25kPa. They already found some differences between 0.1 and 0.4 kPa (both really soft ECM). It will be interesting to determine whether the effect of ECM stiffness on cell metabolism is linear or depend on a threshold? Also it will be of interest to determine whether ECM stiffness can also rewire glucose and/or amino acid metabolism in 3D cell culture.

We thank the reviewer for these suggestions. We initially chose 0.1 kPa and 25 kPa to encompass the range of physiological stiffness that may be encountered by epithelial cells. The intermediate stiffness (0.4 kPa) was chosen since it represents the approximate stiffness of 100% basement membrane extract that we used in 3D cultures. We have repeated these experiments and included two additional stiffnesses, 0.8 and 6.4 kPa. Interestingly, glucose-dependency is linear between 0.1 and 0.8 kPa ($y = -0.0007x + 0.577$; $R^2 = 0.999$) and reaches maximal effect at 0.8kPa. On the other hand, glutamine dependency does not show a linear relationship with stiffness ($y = -8.549e-6x + 0.417$; $R^2 = 0.21$) and varied from 0.1 to 25 kPa. This data is presented as a new Figure 7a.

We also evaluated the effect of stiffness in 3D culture on glucose and glutamine dependency. For this, we embedded cells in Geltrex mixed with 0.2% alginate. We found that in 3D there was a significant requirement for glucose and glutamine under these conditions. We attempted higher concentrations of alginate to determine if there was a linear relationship, however Caco2 cells did not grow well at higher alginate concentrations, even in control medium conditions. Nonetheless, our new data demonstrate that stiffness influences metabolic dependencies in 3D as well as 2D conditions. This is presented as new Figure S8a.

2. As stated by the authors, cancer cells are known to be highly glycolytic compared to non-transformed cells. However, the authors shown that it is strongly dependent on how the cells are cultivated (2D vs 3D). Therefore, it will be of interest to determine: - whether non transformed cells present the same phenotype? - Whether cancer cells still present higher glycolytic rate than non-transformed cells in 3D cell culture?

We thank the reviewer for this suggestion. We examined glucose and glutamine dependency between 2D and 3D environments for MCF10A, a non-transformed mammary epithelial cell line. New data presented in Figure S1(j and k) demonstrate that MCF10A cells display glucose dependency in 2D but not 3D culture, similar to the cancer cell lines we tested. Furthermore, data presented in Figure S6(g and h) also show that MCF10A cells are dependent on glutamine in 2D culture, but not 3D culture. We found similar results for MDCK cells, a non-transformed dog kidney epithelial cell line (Figure below for reviewers only). This data further supports that non-transformed cells show differences in glucose and glutamine dependency between 2D and 3D cultures; we decided not to include the MDCK data in the manuscript for space considerations and for clarity using only human cell lines. These results

support our conclusions that the architectural environment influences glucose dependency and further show that this occurs in both cancer and non-transformed cell types.

Figure 1 for Reviewers: MDCK (dog kidney epithelial cells) were cultured in flat/2D or 3D organotypic cultures for 7 days in complete medium or medium depleted of glucose (Glc-) or glutamine (Gln-). The size (cross-sectional area) was measured for organotypic cultures and normalized to control (complete medium). Cells in flat/2D cultures were trypsinized and counted.

3) To investigate metabolic changes the authors performed Seahorse experiments in 2 cell culture, but also in 3D cell culture. An extended experimental procedure describing how measure OCR and ECAR in 3D cell culture using Seahorse should be provided. At least for this reviewer it appears really hard to understand the technical procedure. Numerous technical caveats have to be lifted (really small distance between the plate and the probe; non confluent area...)

We appreciate the comment and apologize for not being more clear in our methods description for this aspect of our work. We took advantage of the ability to culture Caco2 cells on a thin solid layer of BME with soluble BME in the liquid growth medium that covers the cells. This 3D format has the advantage that it is somewhat similar to the 2D environment, and contrasts other types of 3D culture where cells are fully embedded in a solid gel matrix or are cultured in suspension and require special microwell inserts to capture and immobilize spheroids. At the timepoint we performed Seahorse analysis, 2D/flat cells are less than 70% confluent. Cells in organotypic culture form discrete 3D structures and do not reach confluence like 2D cultures. At the time of Seahorse analysis, 3D structures are approximately 50um in diameter and in log-phase growth. Therefore, these fit under the probe, which is 200um above the bottom of the well. Moreover, we used Caco2 cells because they form a single layer 3D structures, which mitigates potential technical challenges of analyzing solid spheres that likely have diffusion gradients based on distance from the spheroid surface. Therefore, the 3D system we employed is most similar to 2D culture in layout and we used the same

experimental setup. 2D and 3D cultures were run in parallel wells of the same XF24 plates and used the same parameters for OCR and ECAR measurements. Because there are some differences in cell number, we normalized results to cell number. We have also normalized data to baseline readings (i.e. prior to oligomycin treatment) and found that it does not affect the outcome. Therefore, the conclusions are independent of normalization method. We have clarified the culture conditions in revised Fig. 1A and updated the methods to better describe the experimental setup for OCR and ECAR between flat/2D and organotypic/3D cultures as follows:

Metabolic Analyses

The Seahorse XF Mito Stress test (Agilent Cat# 103015-100) and the Mito Fuel Flex test (Agilent, 103260-100) were performed using the XF24 Extracellular Flux Analyzer (Seahorse Bioscience) according to the manufacture's protocols. For both flat (2D) and organotypic (3D) cultures, cells were seeded in XF24 microplates (Agilent Cat# 100777-004). For organotypic cultures, wells were pre-coated with a thin layer of 100% Geltrex, and 2% Geltrex was added to the liquid culture medium, similar to standard growth conditions described above. Cells were cultured for 5 days, during which cells are in log-phase growth. At this timepoint flat/2D cultures form colonies and are <70% confluent; spheroids are approximately 50mm and therefore readily fit under the probe, which is located 200mm above the bottom of the well. Since cells in both 2D and 3D cultures are covered by liquid medium, special spheroid plates were not necessary for 3D cultures and the same XF24 microplates were used for all experimental conditions. On the day of the measurement, cells were washed with XF Base Medium (Seahorse Bioscience Cat# 102353-100) supplemented with D-glucose (25.0 mM) (Sigma Cat# G6152), L-glutamine (4.0 mM) (Bishop Cat# GLU102) and sodium pyruvate (1.0 mM) (Wisent Cat# 600-100-EL). Cell were then incubated in a CO₂-free incubator at 37°C for 1 hour to establish equilibration. For the Mito Stress test, measurements of the oxygen consumption rate (OCR) and extracellular acidification rate (ECAR), were taken before and after the addition of the ATP Synthase inhibitor oligomycin (1 μM), FCCP (0.5 μM) which uncouples oxygen consumption from ATP production, and Rotenone/antimycin A (0.5 μM), which inhibit Complex I and III respectively. For the Mito Fuel test, OCR and ECAR measurements, were taken before and after the addition of the mitochondrial pyruvate carrier inhibitor UK5099 (2 μM), carnitine palmitoyltransferase 1A inhibitor etomoxir (4 μM), or glutaminase inhibitor BPTES (3 μM). For experiments comparing 2D and 3D environments, samples were run in parallel wells on the same plate. In both tests, following ECAR and OCR analyses, cells were trypsinized for cell count and data normalization. ECAR and OCR tracings, glycolytic reserve, maximum respiration, ATP production, proton leak, spare respiratory capacity, coupling efficiency, and non-mitochondrial respiration were calculated from OCR data using Seahorse Wave software (Seahorse Biosciences) and Excel (Microsoft). For both flat (2D) and organotypic (3D), Caco-2 cells were treated for 24 hours with the porcupine/WNT pathway inhibitor, IWP2 (25 μM) (STEMCELL Technologies Cat# 72122) prior to the procedures of the Seahorse XF Mito Stress test described above.

4) The authors performed a GC/MS Metabolic Analysis to compare the metabolic profile of 2D vs 3D cell cultures. It is difficult to understand/interpret their results as the method of normalization remain enigmatic. Also the authors do not provide the full list of metabolites analyzed. Finally, in open science era, it will be of interest that the authors share their raw data.

We have included an updated description of the normalization method used in our study in the methods sections as follows:

GC/MS Metabolic Analysis

Metabolic profiling was performed as previously described⁵⁸. Briefly, cells were cultured in DMEM in 6-well dishes (35mm) for 5 days. Cells were rinsed in saline, quenched in 80% HPLC-grade methanol, sonicated, centrifuged and supernatants were dried in a cold trap (Labconco) overnight at -1°C . Pellets were solubilized in methoxyamine HCl, incubated at room temperature for 1 hour and derivatized with MTBSTFA at 70°C for 1 hour. Next, $1\ \mu\text{L}$ was injected into an Agilent 5975C GC/MS in SCAN mode and analyzed using Masshunter software (Agilent Technologies). Raw GC/MS data were converted to CDF format using Agilent MassHunter Workstation software with the following parameters: Max RT difference between peaks: 0.1 min and Ionization Type: EI. Metabolites were identified by automated comparison of the ion features in the experimental samples to a reference library of chemical standard entries that included retention time and molecular weight (m/z). Peak intensities were quantified using area-under-the-curve (AUC) for all samples. The mean AUC of each metabolite was calculated from 5 technical replicates to yield a single data point for each metabolite in flat/2D and organotypic/3D samples. The fold change was calculated as:

$$\text{fold change} = \frac{\overline{AUC}_{3D}}{\overline{AUC}_{2D}}$$

This was repeated for 3 independent experimental replicates and plotted as mean \pm standard deviation.

As suggested, we have also included triplicate metabolomic data for amino acids and TCA components in Supplementary Table S2. Values reported are area under the curve (AUC) normalized to cell number for each metabolite for which retention times were manually validated.

5) The uptake rates are wrongly calculated as the authors state “Glucose consumption was calculated using the formula $\text{Consumption}_G = [\text{Glc}]_{\text{sample}} - [\text{Glc}]_{\text{blank}}$, where blank wells contained medium, but no cells. Values were then normalized to cell number in each well.” This calculation is only correct in a no growth condition, but the author state their cells grow exponentially. For exponential growth, the calculation should read:

(The equation has been sent to the editor in a pdf file. Please confer with the editor to recover the pdf file.)

Please note Δ refers to the difference of a parameter between two time points.

We thank the reviewer for catching this error. We have corrected the calculations and updated figures (2a and b) and the methods section accordingly.

6) Finally, guided by their RNA sequencing analyses the authors investigate whether Wnt signaling can influence stiffness-dependent metabolic rewiring. The data presented here are only correlative and only rely on the use of a pharmacological inhibitor. Much more work (LOF/GOF and rescue experiments) have to be done to demonstrate the involvement of Wnt signalling pathway in this process.

We agree that the data presented in our manuscript are mostly correlative. We feel an extensive characterization of Wnt signaling in stiffness-dependent metabolic re-programming is of interest, but beyond the scope of the current manuscript. We have revised the text to better reflect the correlative nature of the connection to Wnt signaling.

Reviewer #2 (Remarks to the Author):

1. The manuscript is clearly written and structure and it is accessible. The figure legends tend to contain only a minimum of information. The readability of the figures would be enhanced if some more details were included. A couple more specific comments below relate to this. Some detail is missing from the methods section, a few more specific comments are included below. For some of the equipment that was used, the authors provide only minimal information. Details on important instrument settings are missing.

We thank the reviewer for his/her comments. We have updated the figure legends to provide more information as well as the methods section with more specific information. Also see response to Reviewer 1, question 3.

2. CaCo-2 cells were derived from colon adenocarcinoma and harbor mutations in APC and p53 and other genes, but not the MAP-kinase pathway. It was shown that p53 has inhibitory effects on glycolysis through TIGAR and glucose transporters. In this study glycolysis was only enhanced in the 2-dimensional culture but not in the organotypical cultures. To what extent would p53 deletion be effective in the context of 2D cultures versus 3D spheroids?

This is an interesting question. Although we did not evaluate p53 loss and effects on glycolysis, we did over-express KRAS^{G12V}, which induces glycolysis and promoted glucose-dependency in 3D culture. Since 2D cultures are dependent on glycolysis in the cell lines we examined, we anticipate that 3D systems may be more sensitive than 2D cultures to effects of other genetic alterations like p53 loss on glycolysis. The cell lines evaluated in our study have diverse mutations, including p53 (Caco-2 and OV90) and KRAS (A549), yet we observe a similar trend with glucose and glutamine dependency. Moreover, we have included new data using non-transformed MCF10A cells, which do not have p53 or other mutations in driver oncogenes/tumor suppressors, and we observed a similar effect of glucose and glutamine dependency in 2D growth conditions. This dependence supports our conclusion that the cellular architecture plays a key role in regulating metabolic plasticity independent of genetic alterations.

3. For the Caco-2 cells stably expressing GFP-KRAS G12V, evidence should be shown to what extent the fusion protein was over-expressed.

We have included a blot showing expression of FLAG-tagged KRAS^{G12V} in Caco2 cells is presented in Fig. S7a. GFP was used as a control.

4. Regarding the fluorescent glucose uptake experiment shown in figure 2, example images for start and endpoints for both conditions should be provided to support the quantification data.

Since multiple cell lines were used, the readability of the figures would be enhanced if the different cell lines were clearly indicated per panel.

We thank the reviewer for these suggestions. We have added Figure 2c fluorescent images of initial and end points with 2-NBDG. Caco2 cells have been used throughout the manuscript and additional cell lines are presented in the supplementary figures. We have revised the figure legends to reinforce the cell lines used, and added the cell line labels to the figures when more than one cell line is present.

5. Line 130: FDR should be written out the first time it is used.

This has now been corrected.

6. Can the authors comment why in the gene ontology analysis glycolysis didn't show up as enhanced pathway?

This is an interesting question. In our unbiased gene-ontology analysis, we did not find glycolysis gene sets enhanced, which was initially surprising, given the higher glycolytic activity in 2D cells. We have now explored this further by performing Gene Set Enrichment Analysis (GSEA) specifically on the KEGG_GLYCOLYSIS_GLUCONEOGENESIS gene set. While the enrichment score did not reveal a significant difference between 2D and 3D samples across the whole gene set, we further examined genes at the leading edge to identify potential glycolytic genes that were significantly upregulated in 2D versus 3D culture. This included hexokinase, which catalyzes the conversion of glucose to glucose-6-P, the first non-reversible reaction in glycolysis and PFKP, which catalyzes the conversion of fructose-6-P to fructose 1,6-bisphosphate, the rate limiting step of glycolysis. We have added this as new data in Supplementary Fig. S2a. We think that altered expression of these key enzymes may be sufficient to prime 2D cultures for glycolysis. Moreover, despite a much lower level of basal glycolysis in 3D cultures, they retain capacity for glycolysis under stress. For example, we show in Figures 3b, f that when mitochondrial ATP synthesis is blocked with oligomycin, 3D cells access a glycolytic reserve, which appears necessary to support growth under stress (e.g. mitochondrial stress (Fig. 3) or glutamine-deficient conditions (Fig. 5j)). This would indicate that the glycolytic machinery is present in 3D organotypic cultures, but is not fully engaged. Therefore, the altered expression of a few key genes is not sufficient to trigger a statistically significant gene set enrichment score. Therefore, we think that the retention of basal glycolytic capacity allows cells to better adapt to acute nutrient stress, where glycolysis can readily be utilized without requiring extensive transcriptional regulation to adapt.

7. Figure 2F: I only found SLC2A3/GLUT3 labeled, were other glucose transporter gene expression levels analyzed as well?

Due to space, we only labeled the most differentially expressed metabolic genes on the heat map. All of the differentially expressed genes are present in Supplementary Table S1. Since we have evaluated glycolytic genes specifically in Fig. 2, we have now included additional new data showing the expression of all of the glucose transporters we detected in our RNA-Seq data.. This shows that GLUT1, GLUT3, and GLUT14 are significantly higher in 2D versus 3D (log₂-fold change = 1.7-, 2.9-, and

1.4-fold, respectively), whereas GLUT5 is higher in 3D cultures (log₂-fold change = 0.91). We present this as a new figure (S2b) as a heat map for glucose transporters specifically.

8. Line 286: in this sentence, the 90% reduction of growth refers to glucose free conditions, I assume.

This is correct. We have updated the text to clarify this.

9. Line 351: the first sentence is incomplete.

This has been corrected.

10. Line 431: this sentence is broken.

This has been corrected.

11. Line 438: incomplete sentence.

This has been corrected.

12. Figure legends: it is not always clear what *n* and *r* refer to. Small *r* probably means “repeats” and should be spelled out at least once. For small *n*, it should indicate whether it refers to cells, organoids or other.

Correct, “r” refers to independent replicates and *n* refers to the number of items analyzed (cells, 3D spheroids, fields of view, etc). We have clarified what is measured for each “r” and “n” in the figure legends.

13. Legend panel 1A: since this panel is introducing the experimental setup, it would be helpful for the readers to know what percentage of matrix was used for solid and liquid phases.

Thank you for the suggestion. We have added this information directly to Figure 1a and to the figure legend.

14. Line 696: E-cad need to be spelled out. Along the same line, the information on antibodies for E-cadherin, Par6 should to be included in the methods section.

This information has been included.

15. Line 510: this reference was not included.

This reference has now been included.

16. Scale bars: for panels where all images are displayed at the same scale a single scale bar is sufficient.

This has been corrected.

17. What is a organotypic culture with no lumen? Is it just small groups of cells? From panel 1d it appears that maybe one fourth of the structures had no lumen. What fraction of cells would this affect? I expect that in the organoids without lumen, the cells were not polarized. How would that impact the conclusion that it is the epithelial architecture that explains the metabolic differences?

We thank the reviewer for these questions. In our context, organotypic cultures are three-dimensional spheroids of cells, regardless of the presence of a lumen. In Fig. 1d, the organoids are different sizes, but all of the structures shown have a lumen. Caco2 cells derive from a well-differentiated colorectal adenocarcinoma and ~90% of 3D spheroids form a central lumen in our hands. In solid appearing spheres, it is not necessarily the case that cells are not polarized, as they can form multiple microlumen or display inverted polarity. A major advantage of using Caco2 cells that they efficiently form a single layer of cells surrounding a central lumen and therefore we do not have the confounding effect of comparing monolayer 2D cultures to solid 3D cultures. We are not aware of a method to control the formation of solid versus lumen containing spheres without some form of genetic manipulation (e.g. expression of an oncogene) which may affect metabolic dependencies themselves. Indeed, we expressed KRAS^{G12V}, which generates large solid organotypic cultures, and also induced dependency on glycolysis. Our new data using non-transformed cells (MCF10A (Fig. S1j,k)) may provide further insights into the question. For example, MCF10A cells form a lumen, but unlike Caco-2 cells, they do not form tight junctions or membrane polarity along the apicobasal axis (they do however, polarize the internal trafficking machinery). Therefore, we do not think that the ability to polarize or not is the major architectural element driving metabolic differences because cells in 2D culture can also polarize. Moreover, other cell types we used (A549, MCF7, OV90) do not form a lumen in our experimental setup, but show similar metabolic plasticity in 3D organotypic cultures .

Cells in 2D exist in much stiffer environments. We tested the effect of stiffness on glucose and glutamine-dependency. New data presented in Fig. 7a shows a linear relationship between matrix stiffness and glucose dependence in 2D environments (see also response to Reviewer 1 question 1). Moreover, new data presented in supplementary Fig. 8a also indicates that stiffness in 3D organotypic cultures affects adaptability to glucose- and glutamine-deprivation. We did not observe complete rescue of cell growth in glucose- or glutamine-deficient conditions on 2D surfaces as soft or softer than the basement membrane extract we use in 3D organotypic cultures. Our interpretation is that while stiffness contributes, it does not explain the whole effect. The architectural environment differences between 2D and 3D are extensive. For example, in flat/2D cultures, nutrients access cells from the apical or free surface of the cells in culture. In contrast, it is the basement-membrane facing surface that is exposed to the nutrient rich medium. Therefore, there is a fundamental difference in the side of the cell that is in contact with nutrients and a question for future consideration is whether cell surface transporters for glucose and amino acids may be differentially localized in different architectural environments.

18. In the abstract the authors state that cell lines from multiple cancers were analyzed, which they did. However, the main figures focus on Caco-2 cells and the data on the additional cell lines are found in the supplementary material. Moreover, while there are high resolution images of Caco-2 organotypic cultures provided and the cell polarization is shown by immunofluorescence staining, this analysis is missing for the other cell lines. So, it is not clear to what extent these cultures were polarized. I wonder whether it would be more suitable to focus the abstract on Caco-2 cells rather than include various tissues.

This is a fair assessment. We have removed the mention of other tissues from the abstract. Although some studies have compared metabolism (mostly glycolysis) in 2D and 3D formats, one caveat has been comparing solid 3D structures with monolayer 2D structures. A major advantage of the Caco-2 cells is that they form a polarized monolayer in 2D and 3D environments, which allowed us to evaluate the architectural environment independent of solid 3D spheroids where nutrient or oxygen

gradients likely exist. The A549, OV90 cells do not polarize in 2D or 3D organotypic culture and form solid spheroids in 3D. The MCF7 cells partially polarize and form a lumen in <10% of spheroids. The MCF10A cells form a lumen, but do not polarize in the sense of forming tight junctions and an apical membrane. We have included a description of this in the results section:

Of note, these additional cell lines do not form monolayer cysts with a central lumen indicating that this effect can be independent of monolayer status.

19. *The Geltrex matrix is a composition of basement membrane proteins including collagen and laminin. Could the increased amino acid metabolism and part of the plasticity towards amino acid deprivation be explained by the presence of abundant matrix proteins that could potentially be processed by the cell?*

This is a great question and one we have also considered. It is possible that cells cultured in the absence of amino acids in culture medium can acquire these from the extracellular matrix in 3D cultures. There are reports of uptake of soluble serum proteins through pinocytosis (PMID: 31257175, 23665962). A similar process may occur to uptake soluble matrix proteins in the absence of amino acids in the culture medium. However, whether basement membrane/extracellular matrix proteins can be catabolized and used for amino acids is unknown to us. The previous reports indicate that pinocytosis is induced by Ras-signaling. We did not observe upregulation of genes associated with pinocytosis or endocytosis in 3D culture compared to 2D. However, we do not have RNA-Seq data for cells cultured under amino acid-deprived conditions. Macrophages are able to digest and uptake fluorescently-labelled extracellular collagen I (PMID: 29281816), however, it is not known whether this is used for metabolic purposes. We examined the basement membrane by staining for laminin in cells grown in complete medium or medium depleted of glucose, glutamine, or amino acids various amino acid-deprived conditions (Figure 2 for reviewers). With this admittedly crude experimental setup, we did not detect changes in basement membrane staining integrity between conditions. Moreover, cells in 3D culture can be maintained for ~2-weeks, at which time the gel starts to naturally degrade to a point where it no longer supports 3D spheroid growth. However, we did not detect increased degradation of the gel matrix in cultures lacking glucose or amino acids or changes morphology of the cysts/spheroids, which suggests to us that if ECM proteins are being used by cells, it does not compromise the functional integrity of the ECM. Therefore, the most likely scenario seems to be uptake of soluble proteins from the basement membrane extract while proteins assembled into extracellular matrix scaffold are maintained. To formally address this would require metabolic labelling *in vivo* to produce labelled basement membrane extract from tissues and metabolic tracing; this represents a study unto itself beyond the scope of the present manuscript. However, we have included some of these ideas in the Discussion as follows:

Our data show that cells in 3D organotypic cultures can survive when essential and non-essential amino acids are excluded from the culture medium. Previous studies indicate that enhanced pinocytosis in Ras-driven cancer cells can promote uptake of extracellular soluble proteins to support amino acid metabolism during glutamine deprivation^{44, 45}. Since we cultured organotypic cells in protein-rich basement membrane extract, it is possible that this provides a source of soluble proteins that may be pinocytosed to support amino acid metabolism in 3D organotypic cultures. However, our data also demonstrate that access to a glycolytic reserve in organotypic environments restores growth under glutamine-deficient growth conditions, suggesting that diverse mechanisms for metabolic adaptability likely exist in tissues.

Figure 2 for Reviewers: Caco-2 cells were cultured for 7 days in complete medium or medium depleted of glucose (Glc), glutamine (Gln) or amino acids, then fixed and stained for the extracellular matrix protein laminin-5.

Reviewer #3 (Remarks to the Author):

1. Line 280 "The mechanical environment regulates glucose – dependent growth". The author cultured cells on polyacrylamide gels which coated with collagen solution. If you want to prove your hypothesis of substrate stiffness could influence the glucose and glutamine dependency, why don't you use different concentration of Geltrex, which it was used to culture cells?

We thank the reviewer for suggestion. We chose to use Geltrex/alginate mixture instead of varying the Geltrex composition because we were concerned that different Geltrex concentrations would change both the stiffness and the concentration of extracellular matrix proteins, the latter of which could influence signaling independent of stiffness. Therefore, we produced 50% Geltrex and modified the stiffness by adding alginate. These experiments are presented in a new figure (Fig. S8a) and show that glucose- and glutamine-dependent growth is influenced by stiffness in 3D cultures. Please also see response to Reviewer #1 comment 1.

REVIEWERS' COMMENTS:

Reviewer #1 (Remarks to the Author):

In their revised manuscript, the authors have addressed many reviewer's concerns and requests for clarification regarding their conclusions and the experimental procedures. The major novelty of this manuscript is to provide novel evidence of the crucial role of tissue architecture in metabolic plasticity. The revised manuscript contains new data and clarifications that strengthen this hypothesis and greatly improve the study. The authors should be congratulated for their extensive work.

Thomas BERTERO, PhD

Reviewer #2 (Remarks to the Author):

The authors adequately addressed all my questions. However, there are a couple minor issues left that should be resolved.

Minor

Line 127: it should reference figure 2g instead.

Line 129: typo, it is no instead of not.

Line 188: there may be a typo towards the end of the line. The authors potentially meant to refer to "our earlier data".

Fig S7a: EGFP is 27kDa in size, why would it run above 33kDa?

Line 360: the sentence is not finished.

Line 368: the sentence starting at the end of this line is incomplete.

Line 466: in the second part, "Images" appears abandoned.

Line 568: Regarding the matrix shear moduli, it should be indicated whether the values are estimated or whether they were experimentally determined

The figure legends state that standard deviation is displayed by the error bars. I assume that the various plot display the mean values. The author should clarify this either in the statistics section in the methods or in the figure legends.

Line 969 and 973: the IWP2 drug concentration needs a capital-M in μM .

Reviewer #3 (Remarks to the Author):

The authors did a remarkable job to improve the revised manuscript.

Reviewer #1 (Remarks to the Author):

In their revised manuscript, the authors have addressed many reviewer's concerns and requests for clarification regarding their conclusions and the experimental procedures. The major novelty of this manuscript is to provide novel evidence of the crucial role of tissue architecture in metabolic plasticity. The revised manuscript contains new data and clarifications that strengthen this hypothesis and greatly improve the study. The authors should be congratulated for their extensive work.

We thank the reviewer for their positive comments.

Reviewer # 2 (Remarks to the Author)

Minor

Line 127: it should reference figure 2g instead.

This has been corrected.

Line 129: typo, it is no instead of not.

This has been corrected.

Line 188: there may be a typo towards the end of the line. The authors potentially meant to refer to "our earlier data"

This has been corrected.

Fig S7a: EGFP is 27kDa in size, why would it run above 33kDa?

The marker should be 23kDa instead of 33kDa. We have corrected the figure.

Line 360: the sentence is not finished.

This has been corrected.

Line 368: the sentence starting at the end of this line is incomplete.

This has been corrected.

Line 466: in the second part, "Images" appears abandoned.

This has been corrected.

Line 568: Regarding the matrix shear moduli, it should be indicated whether the values are estimated or whether they were experimentally determined.

These were previously experimentally determined. We revised the methods to indicate the method used to measure the stiffness and reference previous work - "Polyacrylamide hydrogel stiffness was

previously characterized using rheometry⁵⁹.”

The figure legends state that standard deviation is displayed by the error bars. I assume that the various plot display the mean values. The author should clarify this either in the statistics section in the methods or in the figure legends.

We have updated the Statistics section to clarify this point – “Results are presented as mean \pm standard deviation.”

Line 969 and 973: the IWP2 drug concentration needs a capital-M in μM .

This has been corrected.

Reviewer #3 (Remarks to the Author):

The authors did a remarkable job to improve the revised manuscript.

We thank the reviewer for their positive comments.